# Single-cell adaptations shape evolutionary transitions to multicellularity in green algae

Charlie K. Cornwallis ●[1] ✉, Maria Svensson-Coelho[1], Markus Lindh[2], Qinyang Li ●[1], Franca Stábile ●[1], Lars-Anders Hansson ●[1] & Karin Rengefors[1]

The evolution of multicellular life has played a pivotal role in shaping biological diversity. However, we know surprisingly little about the natural environmental conditions that favour the formation of multicellular groups. Here we experimentally examine how key environmental factors (predation, nitrogen and water turbulence) combine to influence multicellular group formation in 35 wild unicellular green algae strains (19 Chlorophyta species). All environmental factors induced the formation of multicellular groups (more than four cells), but there was no evidence this was adaptive, as multicellularity (% cells in groups) was not related to population growth rate under any condition. Instead, population growth was related to extracellular matrix (ECM) around single cells and palmelloid formation, a unicellular life-cycle stage where two to four cells are retained within a mother-cell wall after mitosis. ECM production increased with nitrogen levels resulting in more cells being in palmelloids and higher rates of multicellular group formation. Examining the distribution of 332 algae species across 478 lakes monitored over 55 years, showed that ECM and nitrogen availability also predicted patterns of obligate multicellularity in nature. Our results highlight that adaptations of unicellular organisms to cope with environmental challenges may be key to understanding evolutionary routes to multicellular life.

The evolution of multicellularity has profoundly influenced the diversity of life on earth[1–4]. Obligate dependence among cells for reproduction and survival ('obligate multicellularity') has facilitated the evolution of different cell types capable of performing specialized tasks[5–7]. The novel functions and increased size brought about by obligate multicellularity has allowed the colonization of new environments, shaping several of the major branches of the tree of life including algae (green, red and brown), plants, animals and fungi[1,3,6–8]. While multicellularity has clearly enabled new levels of morphological complexity to evolve, it remains unclear what ecological advantages the initial formation of multicellular groups has and why this has occurred in some lineages but not others[3,4].

Unicellular species that form multicellular groups in response to environmental stimuli ('facultative multicellularity') can provide insight into the ecological benefits of multicellularity[9–16]. Studies on diverse facultatively multicellular taxa have led to the predominant view that predation and nutrient availability are primary factors driving the evolution of multicellularity[1,3,17,18]. For example, in green algae, single cells form groups in response to zooplankton and high nitrogen concentrations[19–22]; in *Dictyostelid* slime moulds, single cells aggregate in the presence of nematode predators[23] and food limitation[24,25]; and in yeast multicellular pseudohyphae-like structures emerge in response to macrophage attack[26] and nutrient stress[27,28]. Predation is thought to increase the benefits of multicellularity through a variety of mechanisms, such as reduced ingestion rates[9,10], sedimentation[16] and resistance to digestion[29,30]. Nutrient limitation, on the other hand, can select for and against multicellularity depending on whether cell–cell cooperation helps harvest nutrients from the environment or is required for the maintenance of multicellular groups[1,3].

[1]Department of Biology, Lund University, Lund, Sweden. [2]Swedish Meteorological and Hydrological Institute, Västra Frölunda, Sweden. ✉e-mail: charlie.cornwallis@biol.lu.se

Our current understanding of the environments favouring multicellularity is, however, limited for several reasons. First, multicellularity can reduce replication rates, potentially offsetting the benefits of predator defence and nutrient acquisition under natural conditions[14,31,32]. In the few cases where the fitness pay-offs of multicellularity have been directly measured rather than assumed from rates of multicellular group formation, results have been inconsistent[14,20,33–35]. Therefore it is unclear whether multicellularity is an adaptive response or is a correlated by-product of changes in other traits responding to environmental stimuli, such as cell size or extracellular excretions[36]. Second, the effects of predation and nutrients have largely been examined in isolation and independently from other environmental factors that may modify the costs and benefits of multicellularity[11–15,31]. For example, in aquatic environments, nutrient levels and the success of predators in capturing unicellular versus multicellular prey may be modified by water turbulence[37]. Third, due to the challenges of isolating species from the wild[38], most research has been conducted on single laboratory strains that have become adapted to artificial conditions[39]. Consequently, we lack a thorough understanding of why species in natural populations form multicellular groups[3].

Here we use phylogenetic comparative analyses to examine the ecological conditions that explain multicellularity in green algae distributed across Swedish lakes using a combination of experiments and long-term lake-monitoring data. Green algae of the order *Chlamydomonadales* are an important model system for studying multicellularity[3,12,40,41]. Predation and nutrient levels have repeatedly been shown to induce the formation of multicellular groups in unicellular species, and there are multiple origins of obligately multicellular species, some of which, such as *Volvox* and *Pleodorina*, have differentiated cell types[12,41,42]. We examined 35 unicellular strains (genetically distinct 18S lineages from 19 species that span the *Chlamydomonadales* phylogeny), 33 of which were collected from ten different lakes and two that are commonly used reference strains (Fig. 1 and Supplementary Table 1). Each strain was exposed to varying levels of predation (*Brachionus calyciflorus* previously shown to induce multicellularity[12,31]), nutrients (high versus low nitrate levels) and water turbulence (still versus turbulent). The different environmental variables were manipulated in a factorial design, and responses were monitored over 14 days (-14–28 generations (that is, mitotic cell divisions)[43], Supplementary Tables 2 and 3). Multicellular groups were defined as more than four cells, as the typical asexual life cycle of unicellular algae involves a stage where mother cells divide by mitosis into four daughter cells, known as a palmelloid. We measured the number of single cells, the number of cells in palmelloids (two–four cells) and the number of cells in multicellular groups (more than four cells) before and after exposure to experimental treatments using a high-throughput particle analyser ('FlowCam').

First, we tested which environmental conditions promote multicellular group formation and the extent to which this was explained by past ecological (lake of origin) and evolutionary history (phylogeny). Second, we quantified the fitness pay-offs of multicellularity under different environmental conditions at two levels: the population growth rate of strains under different conditions was used as a measure of adaptation to different environments (referred to as 'population growth rate'), and relative growth rate (growth rate in environment / mean growth across all environments referred to as 'relative growth rate') was used as a measure of how fitness changes with environmental conditions within strains. Third, we tested whether multicellular group formation was explained by changes in cellular morphology (cell size and extracellular matrix (ECM)) in response to environmental conditions. Finally, we examined if there was general support for environmental conditions influencing the evolution of multicellularity across green algae using a long-term database (55 years) of 478 lakes of 332 species classified as unicellular or obligately multicellular by previous literature (Supplementary Table 4). Our experiments, therefore, examine the facultatively multicellular responses of unicellular species, whereas our analyses of long-term data examine evolved differences between unicellular and obligately multicellular species. We analyse data using Bayesian phylogenetic mixed models (BPMM) with posterior modes, 95% credible intervals (CIs) and approximate *p* values (number of iterations where one level is greater or less than the other level divided by the total number of iterations, pMCMC) used to estimate the relationship between traits (full details of analytical approaches are provided in Methods 'Statistical analyses' and in the Code Availability section).

## Results

### The formation of multicellular groups

Consistent with previous research, predation increased the percentage of cells in multicellular groups, referred to here as 'multicellularity' (Fig. 1; multicellularity: predation versus no predation = 2.73 (CI = 0.28 to 4.83), pMCMC = 0.02; Supplementary Table 5). However, turbulence and nitrate levels both had larger effects on multicellularity that overrode the effects of predation (Fig. 1; multicellularity: turbulent vs still = 3.81 (CI = 1.74 to 6.36), pMCMC = 0.002; high versus low nitrate = 6.73 (CI = 4.46 to 9.48), pMCMC = 0.001; Supplementary Table 5). Under still conditions, the percentage of cells in multicellular groups was low cancelling out the effects of predation, whereas in turbulent conditions predation was significantly positively related to multicellularity (Fig. 1 and Supplementary Table 5). Similarly, low nitrate conditions cancelled out the effects of predation on multicellularity and high nitrate conditions promoted multicellularity (Fig. 1 and Supplementary Table 5).

Collating data across all environmental treatments showed that nearly all strains (91%) formed multicellular groups under at least one environmental condition (Supplementary Tables 1 and 2). However, the percentage of cells contained within multicellular groups was highly variable across strains, and this was related to phylogenetic history (multicellularity: % of variation explained by phylogeny CI = 32.3% (6.98, 54.39); Extended Data Fig. 1 and Supplementary Table 5). In contrast, little variation in rates of multicellularity was explained by the lake that strains were collected from (4.9%, CI = 0.00, 18.31; Supplementary Table 5). The ability of unicellular algae to form multicellular groups was therefore widespread, but phylogenetic lineages varied in the percentage of cells in multicellular groups when exposed to different environmental factors.

### The fitness pay-offs of multicellularity

The population growth of strains was significantly reduced by predation and turbulence (Fig. 2; population growth rate: predation versus no predation = −0.04 (CI = −0.06 to −0.02), pMCMC = 0.001, turbulent vs still = −0.04 (CI = −0.05 to −0.01), pMCMC = 0.001; Extended Data Fig. 2 and Supplementary Table 6). Contrary to the idea that multicellularity offers protection against predators, the percentage of cells in multicellular groups had little effect on the growth of populations when exposed to predators. In fact, multicellularity was not related to population growth rate under any of the environmental conditions (Fig. 2 and Supplementary Table 6). Similar results were found when analysing relative growth rates, further demonstrating that changes in multicellularity within strains had minimal effects on fitness (Extended Data Fig. 3 and Supplementary Table 7).

One explanation for why we failed to detect any anti-predator benefits of multicellularity was that the density of predators was either too high or too low. To investigate this, we conducted a second experiment where the same strains were exposed to different densities of predators (control = 0 per ml, low = about 15 individuals per ml and high = about 50 individuals per ml). Once again, we found that the percentage of cells in multicellular groups did not relate to population growth rates, even though high-predation levels reduced the population growth of all strains (Fig. 2 and Supplementary Table 8). We also examined if strains with a higher prevalence of multicellular groups reduced the population growth of predators, as would be expected if

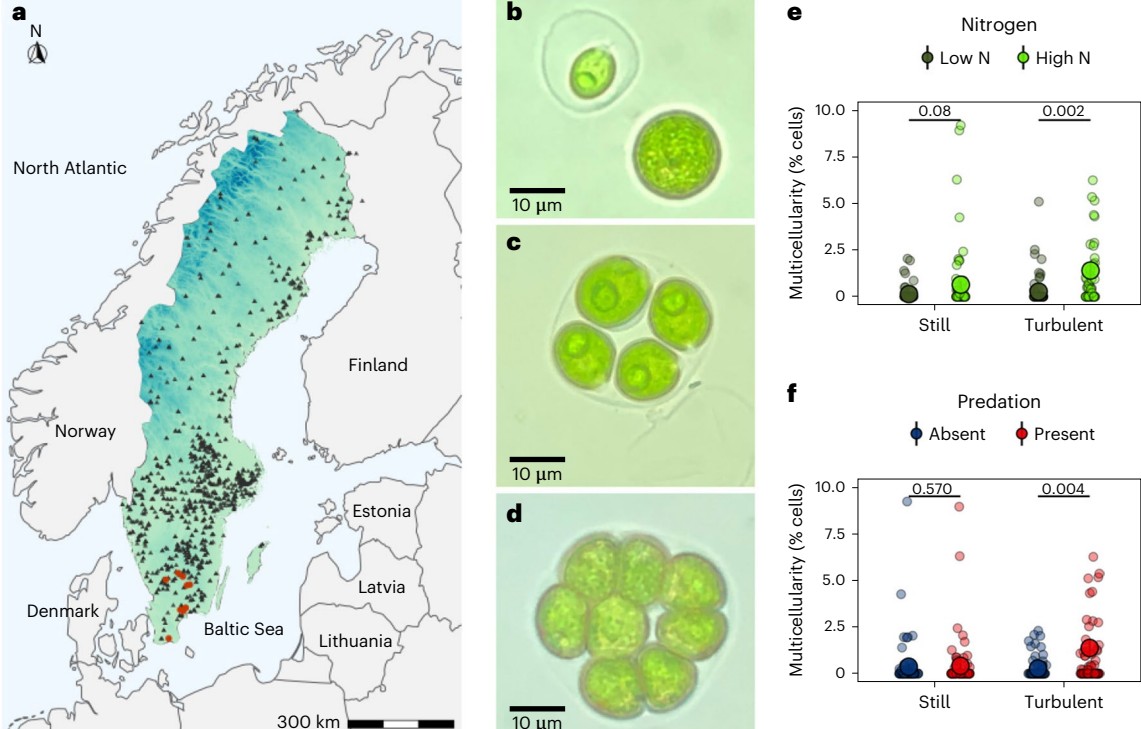

**Fig. 1 | Multicellular group formation across wild strains of green algae depends on water turbulence, predation and nitrate levels. a**, Unicellular green algae were isolated from lakes across southern Sweden. Red dots show the lakes where strains used in experiments were collected, and black triangles show the 478 monitored lakes. **b**, Unicellular species with and without ECM. **c**, Four cells retained in a palmelloid after cell division. **d**, A multicellular group formed by a unicellular species in response to environmental conditions. **e,f**, Strains formed multicellular groups in response to experimental manipulation of environmental

conditions: predator presence, nitrate levels and water turbulence. When exposed to more turbulent water, higher levels of nitrate and *Brachionus* predators, higher percentages of cells were found in multicellular groups. Lines indicate comparisons with pMCMC values from Bayesian phylogenetic mixed models (Methods 'Statistical analyses' and Supplementary Table 5). Small points represent values for each of the 35 strains examined across predation, nitrogen and turbulence treatments (274 data points). Large points with error bars (±1 standard error of the mean (SEM)) are the overall treatment means.

multicellularity reduced ingestion rates or digestibility. The growth of predator populations was not influenced by the percentage of algal cells in multicellular groups, but predator populations that started at a higher density grew more slowly (Supplementary Table 9). Therefore, the fitness of both algal and predator populations were influenced by predator density but were not related to rates of multicellularity.

**Multicellular groups form via the retention of daughter cells**

The apparent lack of benefits of multicellularity in our experiments raises the question of why multicellular groups readily form under specific environmental conditions (Fig. 1). Comparative genomic studies have highlighted that the evolution of multicellularity involves the co-option of genes regulating cell replication in green algae[44,45]. Research on unicellular algae has also shown that the cell-replication cycle can be arrested when mother cells divide mitotically into four daughter cells (palmelloids) in the face of various environmental stressors and when zygospores cannot form, for example, when nitrogen is not limited[13,46]. It is therefore possible that environmental conditions that induce cellular stress alter cell-replication cycles, causing multicellular groups to form as a by-product.

We investigated this idea by examining the relationship between palmelloid formation and multicellularity. Across all environmental conditions, there was a strong positive relationship between the percentage of cells in palmelloids and the percentage of cells in multicellular groups (Fig. 3 and Supplementary Table 10). Additionally, there were higher percentages of cells in palmelloids under turbulent and high-nitrate conditions, similar to patterns of multicellularity (Fig. 3; % cells in palmelloids: high versus low nitrate = 0.67 (CI = 0.42

to 0.96), pMCMC = 0.001, turbulent vs still = 0.35 (CI = 0.14 to 0.66), pMCMC = 0.012; Supplementary Table 11). Conversely, predation did not increase rates of palmelloid formation and even decreased them under still, low nitrate conditions (% cells in palmelloids: predation versus no predation = −0.89 (CI = −1.66 to −0.19), pMCMC = 0.014; Supplementary Table 11).

**ECM promotes palmelloid persistence**

The link between palmelloids and larger multicellular groups was supported by image analysis. It was often observed that groups consisted of numerous daughter cells within a mother-cell wall and multiple palmelloids held together by ECM (Fig. 1). In obligately multicellular organisms, ECM has repeatedly been highlighted as playing an important role in processes such as cell–cell adhesion, cell signalling and nutrient storage[16,47–49]. However, the influence of ECM on the initial formation of multicellular groups is not well established[50,51].

We tested if ECM explained variation in the retention of cells within palmelloids. There was marked variation in the production of ECM. Out of the 35 strains, 28 (80%) produced ECM, and across these 28, the percentage of single cells with ECM varied across environments (range = 0% to 90%; Supplementary Table 2). ECM around single cells strongly predicted the presence of ECM around palmelloids (% palmelloids with ECM: % single cells with ECM CI = 0.81 (0.24, 1.45), pMCMC = 0.01; Supplementary Table 12), and strains that produced ECM had higher percentages of cells in palmelloids across all environmental treatments (Fig. 3; % cells in palmelloids: ECM versus no ECM CI = 0.71 (0.37, 1.1), pMCMC = 0.001; Supplementary Table 11). The production of ECM around palmelloids increased with high nitrate levels

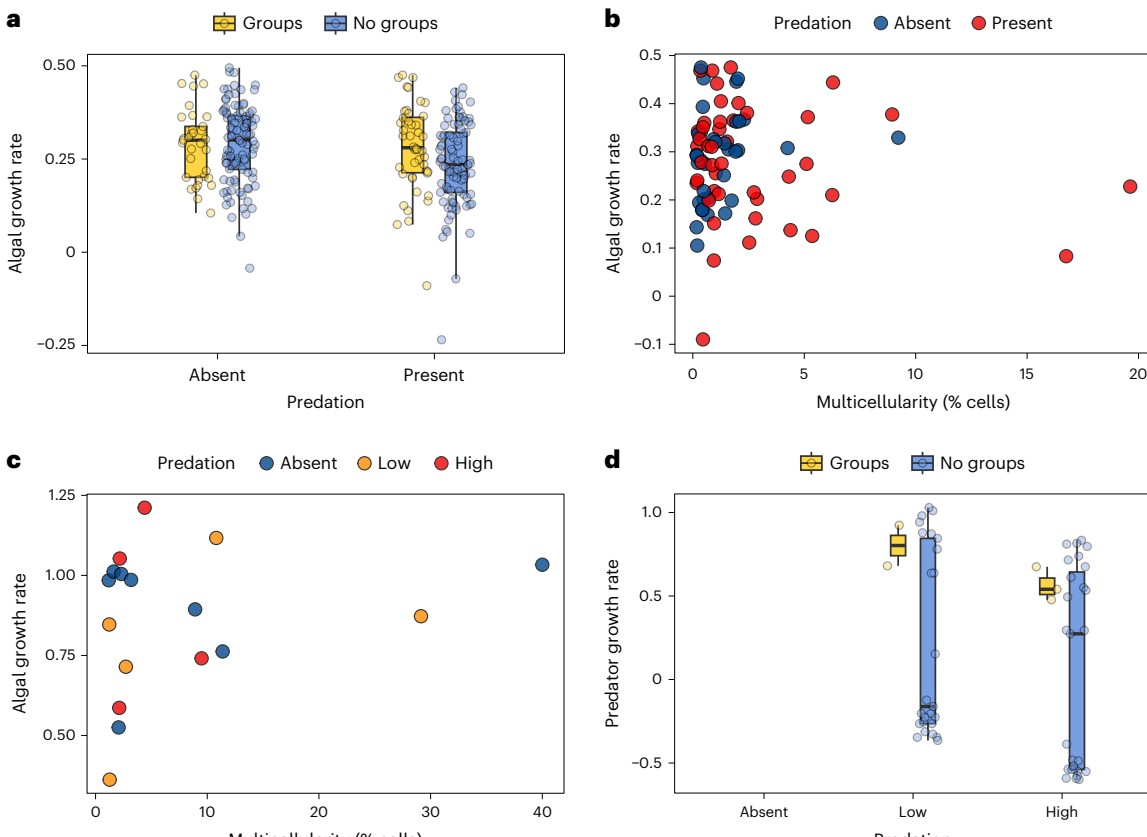

**Fig. 2 | Multicellularity was not associated with fitness under any experimental conditions. a,b,** The presence of multicellular groups (**a**) and the percentage of cells in multicellular groups (**b**) were not related to algal-population growth rates in the absence or presence of predators. **c,d,** The percentage of cells in multicellular groups among strains that formed groups was not related to algal-population growth rates when predation pressure was manipulated (**c**) and did not reduce the population growth of predators (**d**). In **d**, predators were unable to survive on some strains of the *Microglena* clade, generating a bimodal distribution of predator-population growth rates in the

no-multicellular-group category. We suspect this is either because they produce toxins or provide limited nutrients for *Brachionus*. Sample sizes across panels: 35 strains examined across predation, nitrogen and turbulence treatments = 274 data points (**a**); 28 strains formed groups examined across predation, nitrogen and turbulence treatments = 79 data points (**b**); 10 strains formed groups across control, low and high treatments = 17 data points (**c**) and 34 strains examined across low- and high-predation treatments = 57 data points (**d**). The box plots show the median as the central line, the 25th and 75th percentiles as hinges and the extreme values within 1.5× the interquartile range as the whiskers.

(Fig. 3; % palmelloids with ECM: high versus low nitrate CI = 0.84 (0.15, 1.96), pMCMC = 0.016; Supplementary Table 13) but was unaffected by predation and turbulence (Supplementary Tables 12 and 13).

These results suggest that ECM production increases the number of cells retained within palmelloids and, in turn, the formation of multicellular groups. Without manipulating ECM and the number of cells retained within palmelloids, it is difficult to establish if such relationships are causal. Path analysis can, however, be used to evaluate evidence for the most likely causal structure between sets of related variables. We therefore conducted a phylogenetic path analysis to evaluate support for nine models with different causal relationships between ECM, palmelloids and multicellularity (Extended Data Fig. 4a). We found that the best-supported model was one where ECM caused an increase in the percentage of cells in palmelloids, and palmelloids caused an increase in the percentage of cells in multicellular groups. There was also evidence that palmelloid formation may increase the presence of ECM (Extended Data Fig. 4b,c). All other models were rejected (Extended Data Fig. 4).

### The fitness pay-offs of ECM and palmelloid formation

Given the potential effects of ECM and palmelloids on multicellularity, we tested if they influenced population growth rates across different experimental conditions. We found that strains that produced ECM had lower population growth than strains without, particularly under low nitrate conditions (Fig. 4; population growth rate: ECM versus no ECM under low nitrate = −0.14 (CI = −0.23 to −0.04), pMCMC = 0.008; Supplementary Table 14). After controlling for the effects of ECM, rates of palmelloid formation were positively related to population growth rates, especially when predators were present (population growth rate: % cells in palmelloids with predators CI = 0.02 (0.01, 0.05), pMCMC = 0.008; Supplementary Table 14). Given the effects of ECM and palmelloids on population growth and multicellularity, we re-examined the relationship between multicellularity and population growth after controlling for ECM and palmelloids and again found no relationship. Variation in ECM and palmelloid retention were also related to phylogenetic history, indicating that differences between strains may be the result of evolutionary adaptations to different environmental conditions (% of variation explained by phylogeny CI: ECM = 48.45 (27.35, 74.51), palmelloids = 18.79 (6.7, 34.32); Supplementary Tables 11 and 12). There was, however, little effect of ECM and palmelloid formation on relative growth rates (Supplementary Table 15), suggesting that ongoing selection on variation in these traits within strains is weak.

### Ecological conditions explain multicellularity in nature

Finally, we examined the broader importance of our experimental results using a long-term database of phytoplankton abundances and

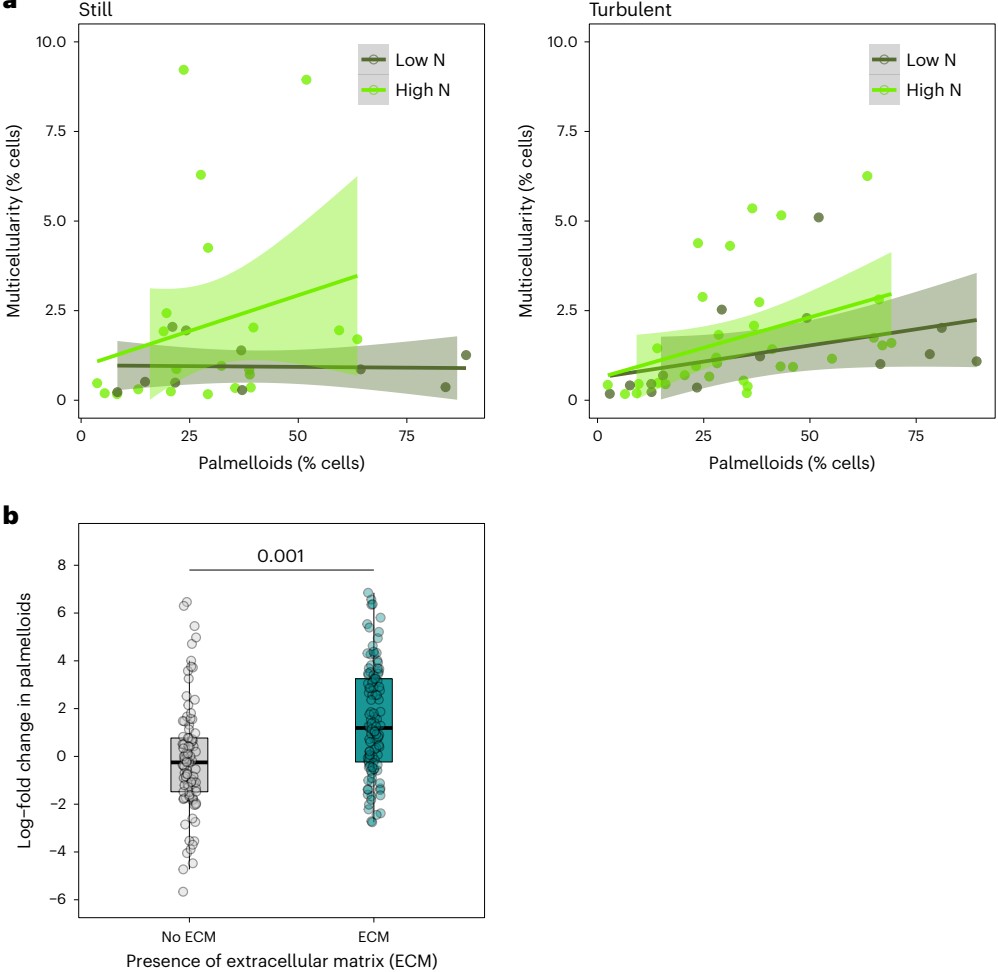

**Fig. 3 | Multicellular group formation is associated with the retention of reproductive cells (palmelloids) surrounded by ECM. a**, In environments with higher levels of nitrate with more turbulent water, higher percentages of cells were found in palmelloids that was positively related to the percentages of cells in multicellular groups. Points represent values for each of the 28 strains that formed multicellular groups across predation, nitrogen and turbulence treatments = 79 data points. Regression lines are plotted with 95% confidence intervals as shaded areas. **b**, The percentage of cells retained within palmelloids over time (log2(% cells at day 0 / % cells at day 14)) was higher in strains that produced ECM. Line indicates comparisons with pMCMC values from Bayesian phylogenetic mixed models (Methods 'Statistical analyses' and Supplementary Table 11). Points are values for each of the 35 strains examined across predation, nitrogen and turbulence treatments = 274 data points. The box plot shows the median as the central line, the 25th and 75th percentiles as hinges and the extreme values within 1.5× the interquartile range as the whiskers.

lake chemistry across Swedish lakes. Using descriptions from the literature of green algae (Chlorophyta), it was possible to determine whether species found in lakes typically produce ECM (80% of species) and are obligately multicellular (58% of species).

We found that unicellular genera that occur in lakes with relatively high levels of dissolved inorganic nitrogen, measured as ammonium, had a higher probability of producing ECM (probability of ECM: log ammonium CI = 139.14 (−64.59, 506.45), pMCMC = 0.076; Supplementary Table 16). Genera with ECM were also more likely to be obligately multicellular than genera without ECM (Fig. 5; probability of multicellularity: ECM versus no ECM = 79.62 (CI = −6.79 to 320.25), pMCMC = 0.036). The strength of the relationship between ECM and multicellularity was, however, mediated by the levels of ammonium in lakes (Fig. 5; probability of multicellularity: log ammonium × ECM versus no ECM = −505.18 (CI = −982.46 to −146.16), pMCMC = 0.001. Supplementary Table 17). Genera without ECM were much more likely to be obligately multicellular if they inhabited lakes with relatively high levels of ammonium, whereas genera with ECM had a high probability of being obligately multicellular regardless of ammonium levels (Fig. 5 and Supplementary Table 17). In contrast to ammonium, there was little effect of nitrite/nitrate levels in lakes on either ECM or multicellularity (Supplementary Tables 16 and 17), possibly due to ammonium being a preferred source of nitrogen for green algae[52].

## Discussion

Multicellular groups formed in response to all environmental variables examined. A key factor related to the formation of larger groups was the retention of cells within palmelloids that were linked to ECM. ECM and rates of palmelloid formation influenced the population growth of strains, whereas multicellularity was not related to population growth under any conditions. It is therefore possible that multicellularity may arise as a by-product of selection on ECM and palmelloid formation. The associations between ECM, nitrogen and multicellularity were broadly evident across a wide range of unicellular and obligately multicellular chlorophyte species in natural lakes. Whether facultative and obligate multicellularity are part of a continuum of sociality or whether they are different evolutionary end points remains unclear; however, the correspondence between the experimental factors that

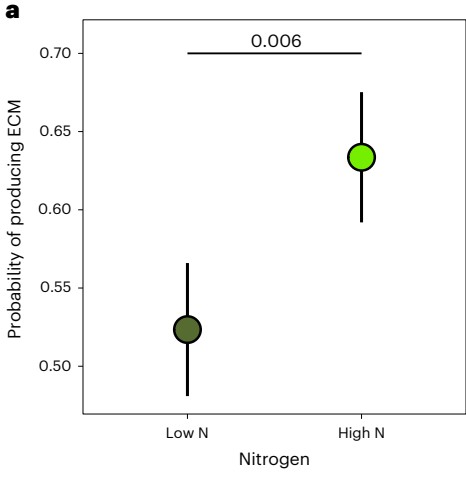

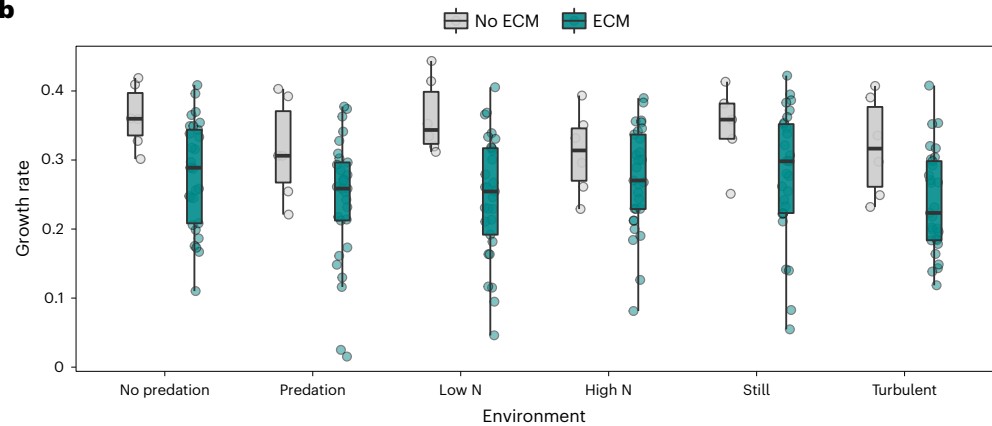

**Fig. 4 | The production of ECM depends on environmental conditions and carries a fitness cost. a**, The probability of producing ECM (>10% cells with ECM) was higher in more nitrate-rich environments. Lines indicate comparisons with pMCMC values from Bayesian phylogenetic mixed models (Methods 'Statistical analyses' and Supplementary Table 12). Means across 35 strains with standard error bars (±1 SEM) are presented. **b**, The population growth of algal strains that produced ECM was significantly lower than those that lacked ECM across all environments (pMCMC = 0.028; Supplementary Table 14), but particularly when nitrate was low (low N pMCMC = 0.008; Supplementary Table 14). Points are mean values for each of the 35 strains. The box plot shows the median as the central line, the 25th and 75th percentiles as hinges and the extreme values within 1.5× the interquartile range as the whiskers.

induced multicellularity and the factors that correlate with obligate multicellularity in natural systems suggests that inferences about the evolution of obligate multicellularity can be made by studying facultatively multicellular species. These results highlight that understanding single-celled response to environmental variation may help explain why certain lineages evolve multicellularly while others do not.

Previous research into the ecological conditions that favour multicellularity has typically focused on the responses of single species to single environmental variables[11–15,31]. By manipulating combinations of environmental variables, our experiments reveal the relative importance of different environmental factors and their interactions for the formation of multicellular groups. Additionally, conducting experiments across a wide range of unicellular species helped expose the relationships between ECM, palmelloid formation and multicellularity that would not have been evident from studying just one species. As ECM, palmelloid formation and multicellularity were not directly manipulated, the causality underlying these relationships remains to be investigated further. While manipulating such traits is challenging, we hope that our results provide the foundational information required to select appropriate species to stimulate work in this direction.

One of our key results was that unicellular green algae with ECM more readily formed multicellular groups. ECM is produced by both unicellular and multicellular organisms in all major eukaryote lineages and has been repeatedly linked to multicellularity[12,48,49,53,54]. ECM is involved in cellular stress responses that can alter the replication rates of unicellular species, a process that may have been co-opted during the evolution of multicellularity[44,55,56]. For example, genetic mutations influencing extracellular excretions in response to stress in unicellular *Chlamydomonas reinhardtii* can lead to the formation of multicellular groups[51]. The release of daughter cells after cell replication in *C. reinhardtii* is also influenced by genes regulating the digestion of ECM and the mother-cell wall (*VLE1*) that have homologues (*VheA*) in the obligately multicellular relative, *Volvox*[53,57]. In addition, the *rls1* gene in *C. reinhardtii* is expressed under stressful conditions and inhibits cell replication. *rls1* has a homologue in *Volvox* called *regA*, which suppresses reproduction in somatic cells, leading to a germ–soma divide[44,58]. This, together with our results, suggests that the production of ECM by unicellular species in response to environmentally induced stress may slow the release of daughter cells after cell replication and that these mechanisms may be co-opted to generate multicellular life cycles.

ECM has been proposed as an adaptation to protect unicellular organisms against various environmental stressors, including predation[29,59] and toxins[59]. We found that turbulence and predation reduced population growth, suggesting they may induce cellular stress. The upregulation of ECM in response to these factors was, however, dependent on nitrogen availability. The ECM of chlorophytes consists

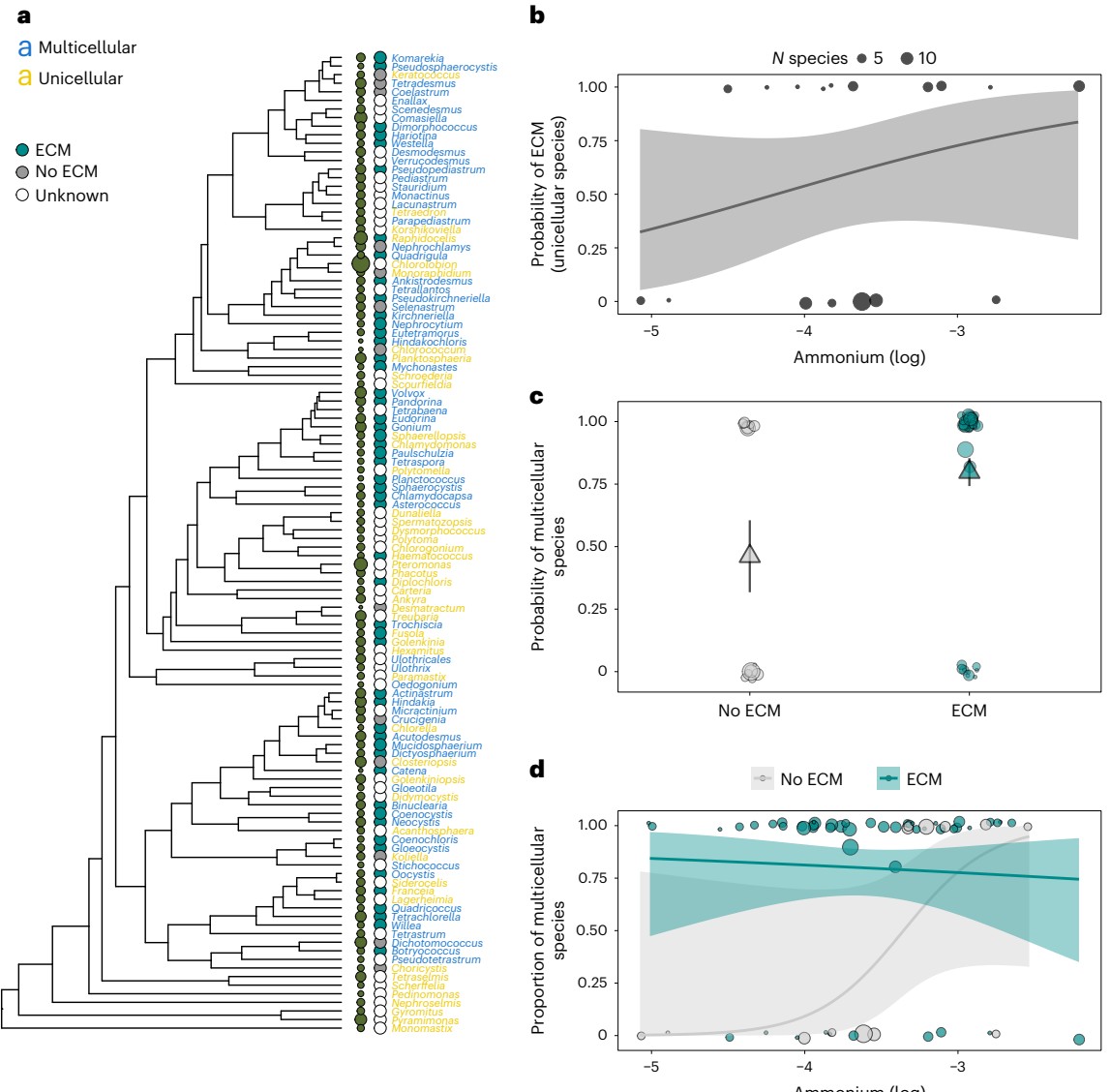

**Fig. 5 | The evolution of obligate multicellularity and ECM is associated with levels of ammonium across Swedish lakes. a**, Phylogeny of multicellular (blue, $N_{genera} = 66$) and unicellular (yellow, $N_{genera} = 48$) genera with (cyan points) and without (grey points) ECM in relation to ammonium concentrations (size of green circles) across 478 lakes in Sweden. A maximum-clade credibility tree of the 1,500 trees sampled for analyses was used for plotting. **b**, The probability that unicellular genera produce ECM increased with the level of ammonium levels in lakes (17 genera with data on ECM; Supplementary Table 4). **c**, Multicellular

genera ($N_{genera} = 47$) had a much higher probability of having ECM than unicellular genera ($N_{genera} = 17$). Large triangles are means across genera with standard errors (±1 SEM). **d**, The probability that genera without ECM ($N_{genera} = 13$) were multicellular increased with ammonium concentrations in lakes, whereas genera with ECM ($N_{genera} = 51$) had a high probability of being multicellular irrespective of ammonium concentrations. In **b**–**d**, round points represent individual genera with sizes proportional to the number of species recorded for each genus. In **b** and **d**, lines are logistic regressions with 95% confidence intervals.

of substantial amounts of nitrogen-rich glycoprotein that may explain the relationships we found between ECM and nitrate in experiments and ECM and ammonium in lakes. Green algae obtain nitrogen more readily from ammonium that inhibits the uptake of nitrate[52,60], which may be why ammonium was correlated to ECM in lakes rather than nitrate (nitrate is used in laboratory media rather than ammonium as it has a greater capacity to buffer pH[61]). ECM and being in a multicellular group may also reduce the ability to absorb nutrients and excrete products that is potentially alleviated in high nitrogen, high turbulence environments.

Alternatively, ECM may be used to store or acquire nitrogen[12], aiding survival through nutrient-poor conditions. We used a standard approach to measure fitness in microbial populations (specific growth rates, for example, ref. 62). However, this is not designed to test for the

effects of nutrient storage, which may be better examined by measuring survival through periods of starvation. It is also possible that the patterns we observed were influenced by the interaction between ECM and nitrogen-fixing bacteria (our cultures were not axenic). ECM can facilitate the capture of bacteria that both fix nitrogen (*Azotobacter chroococcum*[63]) and convert nitrite/nitrate to ammonium (*Methylobacterium* spp.[64]). Bacterial interactions have been proposed to be important for transitions to multicellularity in other organisms[65]. Whether symbioses are also important in modulating the production of ECM in unicellular species in response to environmental conditions, and if this facilitates the evolution of multicellularity, remains an interesting avenue for future research.

Why some lineages have undergone the major evolutionary transition to obligate multicellularity is challenging to explain. Initial

replication costs without the efficiency and task specialization benefits are expected to raise selective barriers to the initial evolution of multicellularity. Our results highlight that the formation of multicellular groups need not necessarily be adaptive but occur through selection on the traits of unicellular organisms[4]. The repeated emergence of multicellular groups, via the plasticity of unicellular organisms, may therefore allow the fitness valleys associated with multicellularity to be traversed[56,66]. If such conditions occur frequently enough, and if conflicts between cells in groups are eliminated (for example, through clonal relatedness between cells[67,68]), then selection may lead to the refinement of adaptive multicellular phenotypes that result in obligate dependence among cells. Examining the evolution of unicellular processes in species with different capacities to form groups may therefore help explain the diverse trajectories to multicellular life[4].

## Methods

### Field collections and culture establishment

Water samples were obtained close to the shore of 20 Swedish lakes (Fig. 1a) in July and August 2016 using a 15 × 50 cm Apstein net with 10 μm mesh size (Hydro–Bios). The samples were examined using an inverted Nikon Eclipse Ts2 microscope at 100× and 200× for single swimming cells matching the general description of *Chlamydomonas* spp.: about 10–15 μm in diameter, two flagella and cup-shaped chloroplast. Single cells were isolated by micropipetting using disposable glass capillaries (Hirschmann Laborgeräte) washed in sterile-filtered lake water and placed in 96-well culture plates (VWR) in 100 μl of a 1:1 mix of filtered lake water and Wright Chu (WC) medium[69] modified by 0.002 mg l$^{-1}$ Na$_2$SeO$_3$•5H$_2$O (MWC+Se). Cultures were maintained at a 12:12 light:dark cycle in 20 °C and 85 μmol photons m$^{-2}$ s$^{-1}$, transferred into larger plates as they grew and finally placed in 25 cm$^2$ non-treated culturing flasks (Thermo Fisher Scientific) containing 30 ml MWC+Se medium.

### Strain identification

Strains were identified to approximate species by Sanger sequencing of the 18S gene and matched to a published gene tree of the order Chlamydomonadales[70]. To sequence cultures, 25 ml was pelleted using a Allegra X-30R Centrifuge (Beckman Coulter) at 3,000× g for 10 min. The supernatant was poured off, and the pellet was gently homogenized by pipetting and transferred to a microcentrifuge tube for centrifuging at 3,000× g for another 10 min. The supernatant was removed with a pipette and the pellet frozen at −80 °C.

**DNA extraction.** The pellet was thawed and frozen five times; 5 min at 37 °C and 30 min at −80 °C. Before the final freezing, 100 μl cetyltrimethylammonium bromide (CTAB) buffer[71] without beta-mercaptoethanol was added to the pellet. DNA was then extracted the next day[71] with some modifications: cells were homogenized in a total of 500 μl CTAB buffer using an electric toothbrush base with a sterile plastic pestle. Pestles were rinsed off into the tube with the cells with 200 μl CTAB buffer. Samples were incubated at 65 °C for 1 hour, vortexing every 15 min. Once cooled to room temperature, the sample was incubated for 20 min with 500 μg RNase A (Qiagen). After adding equal volume of chloroform-isoamyl alcohol (24:1 v/v), samples were rocked at 133 r.p.m. for 20 min before centrifugation. The DNA pellet was eluted with Qiagen EB buffer. DNA quality and concentration were assessed by gel electrophoresis, Qubit 2.0 Fluorometer (Thermo Fisher Scientific) and Nanodrop 2000 v.1.6.198 (Thermo Fisher Scientific).

**18S Polymerase Chain Reaction (PCR).** Geneious v. 11 was used for all sequence processing. We designed primers for amplifying Chlamydomonadales by aligning four phylogenetically distant 18S sequences available on GenBank (*Chlamydomonas monadina*, GenBank ID FR854389; *C. moewusii*, U41174; *C. reinhardtii*, AB511834 and *C. noctigama*, AF008239) and scanning the alignment for conserved regions. A 1,330-base pair (bp) fragment of 18S was amplified in 25 μl

reactions 200 uM of each dNTP, 1.5 mM MgCl$_2$, 400 nM of each primer (18S_ChlamyF 5′ TGCCCTATCAACTTTCGATGGT 3′; 18S_ChlamyR 5′ GTGTGTACAAAGGGCAGGGA 3′), 10 μg bovine serum albumin, 0.5 units AmpliTaq DNA Polymerase (Roche Molecular Systems Inc.) and 1× AmpliTaq buffer. PCR conditions were 96 °C for 5 min followed by 33 cycles of 96 °C for 1 min, 50 °C for 1 min and 72 °C for 3 min and a final extension at 72 °C for 7 min. Products were sequenced in both directions using an Applied Biosystems 3500 Genetic Analyser at the DNA Sequencing Facility, Department of Biology, Lund University. Chromatograms were manually edited and forward and reverse strands aligned. Consensus sequences of each sequence were matched to one another to identify identical haplotypes using the 'De Novo Assemble' function, setting the % maximum mismatches per read to zero. Duplicated haplotypes were excluded from the phylogenetic analysis.

### Phylogeny construction

All 18S sequences used for the published gene tree of ref. 70 were downloaded from GenBank. They were aligned with multiple alignment using fast Fourier transform (MAFFT) using an E-ins-i algorithm with a 1PAM/k = 2 scoring matrix and a gap open penalty of 3. Several species caused gaps in the alignment, and because they were not closely related to our sequences, they were removed and the remaining sequences realigned. Sites were removed from the alignment if they consisted of 95% gaps or data contained identical nucleotides. The remaining 317 18S sequences, covering the full breadth of the order Chlamydomonadales (Chlorophyceae), were then aligned with our 41 haplotypes and three outgroup taxa in the Ulvophyceae (*Ulva compressa*, GenBank ID AB425967; *Ulva californica*, AY303586 and *Ulothrix zonata*, JX491154). Ends of the alignment where only a few sequences had data were deleted, resulting in a final alignment of 360 sequences 1,123 nucleotides long.

The best model of evolution, generalised time reversible (GTR) + proportion of invariable sites (I) + gamma rates (G), was selected in jModeltest2[72] based on the Bayesian information criterion (BIC). We generated an ultrametric gene tree in BEAST v. 1.8.4 (ref. 73) using the best model, an uncorrelated relaxed clock with a log-normal distribution, and the Yule Process of speciation for the tree prior[74]. Runs with 30 million MCMC iterations, sampling every 1,000, were performed through the CIPRES Science Gateway[75]. Appropriate burn-ins were chosen in Tracer v. 1.6 for each run, and the analysis was repeated until we reached an effective sample size greater than 200. In total, trees generated by seven independent runs were combined in LogCombiner after removing 25–40% of iterations as a burn in, and the best majority rule consensus tree was saved.

A phylogenetic tree for the 114 genera recorded in the SLU Miljödata-MVM database, identified by microscope, was constructed by downloading 18S gene data for one species per genus from Genbank (Supplementary Table 4 provides genbank identifiers). Sequences were aligned with three outgroup taxa (*Chara longifolia*, AF032741; *Chlorokybus atmophyticus*, AY823715 and *Mesostigma viride*, KJ808698) in Geneious. A BEAST analysis was performed as explained above. In total, trees generated by six independent runs were combined, removing 10% of iterations as a burnin.

### Preparation of algae cultures for experiments

We selected 33 strains from across different lakes and the phylogeny. Cultures were also prepared for *Chlamydomonas reinhardtii* (K-1016; equivalent to CCAP 11/32 A, SAG 11-32B(89), UTEX 90) and *Edaphochlamy debaryana* (K-1011; unknown isolator) obtained from a culture collection in Copenhagen (Scandinavian Culture Collection for Algae & Protozoa (SCCAP)). We discovered through our 18S sequencing that K-1011 was mislabeled in the SCCAP collection and instead matches *Microglena monadina*. To prepare inocula, 1 ml of each strain were added to 30 ml of MWC + Se media (1:30 ratio) in 50-ml flasks and allowed to reach approximate stationary phase in a GC-300TL climate

chamber (LabCompanion, JeioTech) at 24 °C with shaking using a MR-12 Rocker–Shaker (Biosan) and a 16:8 light:dark cycle at 5,000 lx.

### Experimental design
**Experiment 1 manipulating environmental conditions.** Each strain ($N = 35$) was exposed to different levels of predation, nitrate ($NO_3$) and turbulence in a factorial design consisting of eight treatments: (1) no predation, high nitrate and still (no-turbulence); (2) predation, high nitrate and still; (3) no predation, low nitrate and still; (4) predation, low nitrate and still; (5) no predation, high nitrate and turbulent; (6) predation, high nitrate and turbulent; (7) no predation, low nitrate and turbulent and (8) predation, low nitrate and turbulent. The experiment started by inoculating 30 ml of treatment media with about 5,000–10,000 cells ml$^{-1}$ counted using a FlowCam (Yokogawa Fluid Imaging Technologies, Inc.). The initial number of cells for some strains was variable due to the difficulties in estimating the absolute number of cells in multicellular groups in the inoculum, but this was controlled for in analyses (Methods 'Statistical analyses'). In the predation treatments, *B. calyciflorus* cysts ($n = 50$–100 cysts) were placed in flasks and checked 24 hours later to make sure cysts had hatched. Cysts of *B. calyciflorus* were obtained from tubes in the Acute ROTOXKIT F kit (MicroBioTests Inc.). In high-nitrate treatments, standard MWC + Se medium was used, and in low nitrate treatments, 5% of nitrate was used. To simulate turbulent conditions, flasks were placed on a shaking table with continuous tilting. The experiment was run in batches of five randomly picked strains at a time ($n = 5$ strains × 8 treatments = 40 cultures) for 14–15 days. The experiment ran from December 2017 to February 2018.

The number of cells and the proportion of cells in multicellular groups were measured at the start ($t0$), at 48 hours ($t2$) and at days 14–15 ($t14$) when the cultures reached approximate stationary phase. A sample of 500 µl, diluted when necessary to stay below 500,000 particles ml$^{-1}$, was run through a 100 µm flow cell at ×10 magnification. The output was subsampled to a final particle count of 400. Each of the 400 particles were classified into four categories: single cells, two to four cells typically surrounded by a mother-cell wall (palmelloids) and groups of more than four cells (multicellular groups) using the classify function in VisualSpreadsheet V.3.7.5 (Yokogawa Fluid Imaging Technologies, Inc.). Automated classifications were manually checked to correct classification errors and to count the number of cells in multicellular groups.

**Experiment 2 manipulating predator density.** To infer the effect of predator density on algal-population growth rates, an additional experiment was performed. The same strains as in experiment 1, apart from one culture that crashed in between the two experiments (Har04), were exposed to the following treatments for seven days: (1) no predation, (2) low predation (about 15 individuals ml$^{-1}$) and (3) high predation (about 50 individuals ml$^{-1}$). Algae were transferred from standard culture conditions (MWC + Se medium, still) to a six-well plate at a concentration of about 5,000 cells ml$^{-1}$ in a volume of 5.5 ml and left for 24 hours. The density of *B. calyciflorus* was manipulated by hatching cysts and counting them under a Nikon SMZ1270 stereomicroscope into each well. The number of *B. calyciflorus* at day 7 ($t7$) were counted, either directly in wells if there were few individuals or by taking a subsample of 112 µl and counting in a Palmer–Maloney chamber using an inverted light microscope (Nikon Eclipse Ts2). Aliquots were transferred from experimental wells into 15 ml Falcon tubes before counting, and cultures were stunned with $CO_2$ to facilitate counting of *B. calyciflorus*. Additionally, at $t1$ (24 hours), five individual *B. calyciflorus* per strain from the high treatment were examined at 400× to determine if they had ingested algae. Cell concentrations of algae, the proportion of palmelloids and the proportion of multicellular groups were measured at $t0$ and $t7$ using a FlowCam as described in Methods 'Experiment 1'.

### Estimating fitness pay-offs using population growth rates
Population growth rates of algae and predators were calculated as follows:

$$\text{Specific growth rate per day (SGR)} = \frac{\log_2(N_{t1} - N_{t0})}{t_1 - t_0}$$

where $N_{t0}$ and $N_{t1}$ are concentrations at the beginning and the end of the experiment and $t_0$ and $t_1$ are time in days at the beginning at end of the experiment. Relative specific growth rates (RelSGR) were calculated by dividing SGR by the mean SGR across all experimental conditions.

### Measuring ECM
The proportion of cells and the proportion of palmelloids with ECM was measured by taking the first ten FlowCam images of single cells and palmelloids in every treatment at $t0$ and $t14$. Each image was visually inspected for ECM exudates outside of the cell wall.

### Measuring cell size
The size of individual cells was estimated using the Equivalent Spherical Diameter (ESD) from the FlowCam output. The ESD estimate of cell size represents the full particle, which in the case of cells with ECM can incorporate both the cell and its surrounding ECM layer. To determine if ESD measurements were inflated by ECM, we analysed images generated by the FlowCam in Adobe Photoshop CS6. From the single-cells category at the final time point only ($t14$), we selected the first ten cells with and without ECM. Adobe Photoshop was used to count the number of pixels contained inside the cell wall (that is, ECM was excluded), which was converted to µm$^2$ using a scale bar. We correlated the ESD measurements with Adobe Photoshop measurements for cells with and without ECM. Both were highly correlated (Pearson's correlation: 0.93, 95% confidence interval = 0.81 to 0.81, $p < 0.0001$), indicating that ESD measurements accurately captured the size of cells with and without ECM.

### The environments and phenotypes of green algae in Swedish lakes
Information on water chemistry and phytoplankton biovolumes (mm$^3$ l$^{-1}$) from all lakes in Sweden from years 1964 to 2019 was downloaded from SLU Miljödata-MVM (http://www.slu.se/miljodata-MVM) on 18 January 2019. For each species in the database, we performed literature searches to collect data on multicellularity (a species was considered multicellular when the typical cell number in a group was greater or equal to four) and the presence of ECM, sometimes referred to as mucus or mucilage. The primary reference for all species was 'Växtplanktonflora' (phytoplankton flora)[76]. The second main reference was 'Algaebase' (www.algaebase.org). When these sources did not have information about species, freshwater field guides in Europe or the original publication describing the species were used (Supplementary Table 4[76–112]).

In total, there were 332 species from 114 genera. Multicellularity and ECM presence were typically invariant within genera. Records for some species were sparse and we therefore summarized phenotypic information at the genus level and calculated the median concentration of ammonium, nitrite + nitrate and total phosphorus concentrations of lakes where each genus was found. Total phosphorus was included in analyses as it is a critical macronutrient that is often limiting for algal growth. Species with less than ten occurrence records were excluded from the dataset. The full dataset with references for cell number and ECM is presented in Supplementary Table 4.

### Statistical analyses
**General approach.** Data were analysed in R[113] using Bayesian Phylogenetic mixed models (BPMM) with Markov chain Monte Carlo (MCMC) estimation implemented using the R package 'MCMCglmm'[114]. Default priors were used for fixed effects (independent normal priors

with zero mean and large variance ($10^{10}$)) and inverse gamma priors for random effects ($V = 1$, $v = 0.002$). Random effects were used to model the non-independence of data arising due to multiple data points per strain, multiple strains per lake and the phylogenetic non-independence among strains. Phylogenetic relationships were modelled by fitting a variance–covariance matrix constructed from the phylogeny. To account for uncertainty in phylogenetic relationships, we ran models across a sample of 1,500 trees. Estimates from the last iteration from tree $i$ were used as starting parameter values for tree $i + 1$. Estimates from the last iteration of each tree were saved, with samples from the first 500 trees being discarded as a burn in. For analyses of experimental data, each tree was sampled for 1,000 iterations with a burn in of 999 and a thinning interval of 1. For analyses of the genera found in natural lakes, convergence took longer so each tree was sampled for 10,000 iterations with a burn in of 9,999 and a thinning interval of 1. Model convergence was assessed by running models three times and examining the correspondence between posterior traces, levels of autocorrelation between samples and Gelman and Rubin's convergence diagnostic, where potential scale-reduction factors <1.1 indicate convergence[115].

In Bayesian mixed models, testing the significance of the overall interaction effects (for example, predation × nitrate) is not possible, as it is with frequentist techniques. Only estimates of whether specific combinations of variables differ can be obtained (for example, difference between presence and absence of predators under high nitrate versus low nitrate). Therefore, to quantify the overall contribution of interaction effects to variation in response variables, we used $R^2$ and deviance information criteria (DIC). DIC values were only calculated for models with Gaussian error distributions because there are issues with analytically marginalizing over random effects for non-Gaussian response variables. $R^2$ values were calculated as the square of the correlation between model predictions and raw values of response variable (S. Nakagawa, personal communication). Models with the highest $R^2$, fewest estimated parameters and lowest DIC values were selected and interaction effects were investigated by calculating the differences between combinations of different levels of interacting fixed effects in the model.

Parameter estimates for fixed and random effects are reported as posterior modes with 95% credible intervals (CIs) from models that included all terms of the same order and lower. For example, main-effect estimates are from models where all other main effects are included, estimates of two-way interactions are from models with all two-way interactions and their main effects included and so forth. Parameter estimates from models with binomial error distributions are presented on the logit scale. Fixed effects were considered significant when 95% CIs did not overlap with 0 and pMCMC were less than 0.05. By default, MCMCglmm reports parameter estimates for fixed factors as differences from the global intercept. This does not allow absolute estimates for all factor levels to be estimated, or custom hypothesis tests of differences between factor levels to be performed. We therefore removed the global intercept from models to obtain estimates for each factor level. Differences between factor levels were estimated by subtracting the posterior samples from one level from the comparison level and calculating the posterior mode, 95% CI and pMCMC. All continuous explanatory variables were $z$ transformed before analysis using the 'scale' function in R and explanatory variables that were proportions were logit transformed.

Random-effect estimates presented are from models that included the highest-order interactions of all fixed-effect terms. To estimate the magnitude of random effects, we calculated the percentage of the total random-effect variance explained by each random term on the expected data scale ($I2 = V_i/V_{total}$) (ref. 116). To obtain estimates of I2 on the expected scale from binomial and binary models, the distribution variance for the logit link function was included in the denominator ($V_i/(V_{total}+^{3/2}) \times 100$).

## Specific analyses

**The formation of multicellular groups.** The effect of environmental conditions on the percentage of cells in multicellular groups at the end of the experiment was modelled using a BPMM with a binomial error distribution with number of cells in groups versus number of cells not in groups as the response variable. Three two-level factors, (1) predation (predation versus no predation), (2) nitrate (high versus low) and (3) water turbulence (turbulence versus still) were entered as fixed effects (R code: models 'Grp_1').

**The fitness pay-offs of multicellularity.** The fitness pay-offs of multicellularity were analysed in four ways: (1) to test how differences in multicellularity influence population growth across strains, variation in algal SGRs were modelled using a BPMM with a Gaussian error distribution. Predation, nitrate, water turbulence, the presence of multicellular groups (two-level factor: present versus absent) and the percentage of cells in multicellular groups when multicellular groups were present (specified using the 'at.level' coding in MCMCglmm) were fitted as fixed effects (R code: models 'GR_1'). (2) To test how growth rates vary within strains in relation to changes in multicellularity we re-ran the models in (1) but with relative SGR as the response variable (R code: models 'RelGR_1'). (3) The influence of the percentage of cells in multicellular groups on the population growth of algae under varying intensities of predation were examined using a BPMM with a Gaussian error distribution on data from experiment 2. Predation treatment (three-level factor: no predation, low density and high density), the presence of multicellular groups and the percentage of cells in multicellular groups were fitted as fixed effects (R code: models 'GrazGR_1'). (4) To examine the consequences of variation in multicellularity for predator populations we re-ran the models in (3) but with the population growth rates of predators as the response variable (R code: models 'PredGR_1').

**Multicellularity in relation to palmelloid formation.** To test if the percentage of cells in palmelloids explained variation in the percentage of cells in multicellular groups, we re-ran models outlined in 1 including the percentage of cells in palmelloids as a fixed effect (R code: models 'SP_1').

**ECM and palmelloid persistence.** The effect of environmental conditions and the presence of ECM on palmelloid formation was analysed using a BPMM with a binomial error distribution with number of cells in palmelloids versus the number of single cells as the response variable. Predation, nitrate, water turbulence and the presence of ECM (two-level factor: present versus absent) were fitted as fixed effects (R code: models 'RetainP_1'). We also included the percentage of cells in palmelloids at the start of the experiment as a fixed effect to examine increases and decreases in palmelloid formation in response to environmental conditions.

**ECM production.** To test the effect of environmental conditions on the production of ECM, the number of cells with and without ECM were analysed using a BPMM with a binomial error distribution with predation, nitrate and water turbulence fitted as fixed effects (R code: models 'Mu_1'). Strains where ECM was never observed were excluded from this analysis.

**Path analysis of ECM, palmelloids and multicellularity.** We used phylogenetic path analysis to examine the potential causal relationships between the presence of ECM, the % of cells within palmelloids and the % of cells in multicellular groups using the R package 'phylopath'[117]. Phylogenetic path analysis accounts for the non-independence of data arising from shared evolutionary history by fitting a phylogenetic covariance matrix in regression models. To our knowledge, phylogenetic path analysis has not been

developed to model repeated measurements of species across phylogenies. To account for the non-independence of repeated measurements on the same strains and strains from the same species, we modified the covariance matrix used in analyses to include repeated measurements as polytomies at the tree tips. Specifically, new descendent tips were added to each species in the phylogeny that corresponded to different strains, and descendent tips were added to each strain that represented each observation. This was done using the bind.tip function in the R package 'phytools'[118] with strains being bound to tips and observations being bound to strains. The resulting covariance matrix from the phylogeny was then used in analyses in the way described in the phylopath package (R code provides details). Nine potential causal models were evaluated using data from across all experimental conditions (Extended Data Fig. 4). The relative support for each model was assessed using C-statistic information criterion corrected for small sample sizes (CICc) weights (R code: models 'Path_1.1').

**The fitness pay-offs of ECM and palmelloid formation.** The influence of the presence of ECM and the percentage of cells in palmelloids on the population and relative growth rates was analysed using the same models as described in 'The fitness pay-offs of multicellularity' (1) and (2) with the addition of the presence of ECM and percentage of palmelloids as fixed effects.

**ECM of unicellular taxa and nitrogen in natural lakes.** The relationship between the probability of unicellular genera having ECM and the availability of dissolved inorganic nitrogen, measured as ammonium and nitrate + nitrite concentrations, in the lakes they inhabit was analysed using a BPMM with a binomial error distribution. To account for differences in number of species within genera, we modelled the probability of ECM as the number of species within genera with ECM versus the number of species without ECM. The mean log-transformed concentrations of ammonium and nitrate + nitrite across all samples where each genus was found were fitted as fixed effects.

**Multicellularity, ECM and nitrogen in natural lakes.** We tested if the evolution of multicellularity is related to ECM presence and ammonium and nitrate + nitrite concentrations in lakes using a BPMM with a binomial error distribution. The probability of genera being multicellular was modelled as the number of multicellular species versus number of unicellular species within genera as the response variable. The presence of ECM (ECM presence did not differ between species within each genus) and log-transformed concentrations of ammonium and nitrate + nitrite were fitted as fixed effects.

**Verification analyses.** Our focus was on the effect of nitrogen availability on the presence of ECM and multicellularity. However, the ratio of nitrogen to phosphorus is known to be important for green algae growth. Nitrate + nitrite concentrations were strongly correlated with phosphorus (Pearson's correlation $r = -0.69$) and were therefore not included in the same models to avoid problems with collinearity. We therefore repeated the lake data analyses using the ratio between nitrate + nitrite and total phosphorus. We found no relationship between the probability of unicellular genera having ECM and the ratio between nitrate + nitrite and phosphorus (R code: models 'V1'; Supplementary Table 18), and there was also no relationship between the probability of genera being multicellular and the ratio between nitrate + nitrite and phosphorus (R code: models 'V2'; Supplementary Table 19).

**Reporting summary**

Further information on research design is available in the Nature Portfolio Reporting Summary linked to this article.

## Data availability

Data generated and analysed during this study are available at the Open Science Framework[119] (https://osf.io/b9wpj/). Source data are provided with this paper.

## Code availability

The R code and R session information generated during this study are available at the Open Science Framework[119] (https://osf.io/b9wpj/).

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

## Acknowledgements

We are very grateful to T. Uller for useful discussions and comments on the manuscript, S. Khandan for isolating and setting up cultures, S. B. Gartner for assistance with experiments and S. Nakagawa for statistical advice. We are very grateful for funding from the Knut and Alice Wallenberg Foundation (Wallenberg Academy fellowship: 2013.0129 & 2018.0138 to C.K.C.), Templeton Foundation (60501 to C.K.C.), Crafoord foundation (20210788 to C.K.C.) and Swedish Research Council (VR) (2022-03503 to C.K.C. and 2016-03552 to L.-A.H.).

## Author contributions

Conceptualization: C.K.C., M.S.-C., M.L., L.-A.H., K.R. Methodology: C.K.C., M.S.-C., M.L., Q.L., F.S., L.-A.H., K.R. Investigation: C.K.C., M.S.-C., M.L., Q.L., F.S., L.-A.H., K.R. Visualization: C.K.C., M.S.-C., M.L., Q.L., F.S., L.-A.H., K.R. Funding acquisition: C.K.C. Project administration: C.K.C., M.S.-C. Supervision: C.K.C., M.S.-C. Writing, original draft: C.K.C., M.S.-C. Writing, review and editing: C.K.C., M.S.-C., M.L., Q.L., F.S., L.-A.H., K.R.

## Funding

## Competing interests

The authors declare no competing interests.

## Additional information

**Extended data** is available for this paper at https://doi.org/10.1038/s41559-023-02044-6.

**Correspondence and requests for materials** should be addressed to Charlie K. Cornwallis.

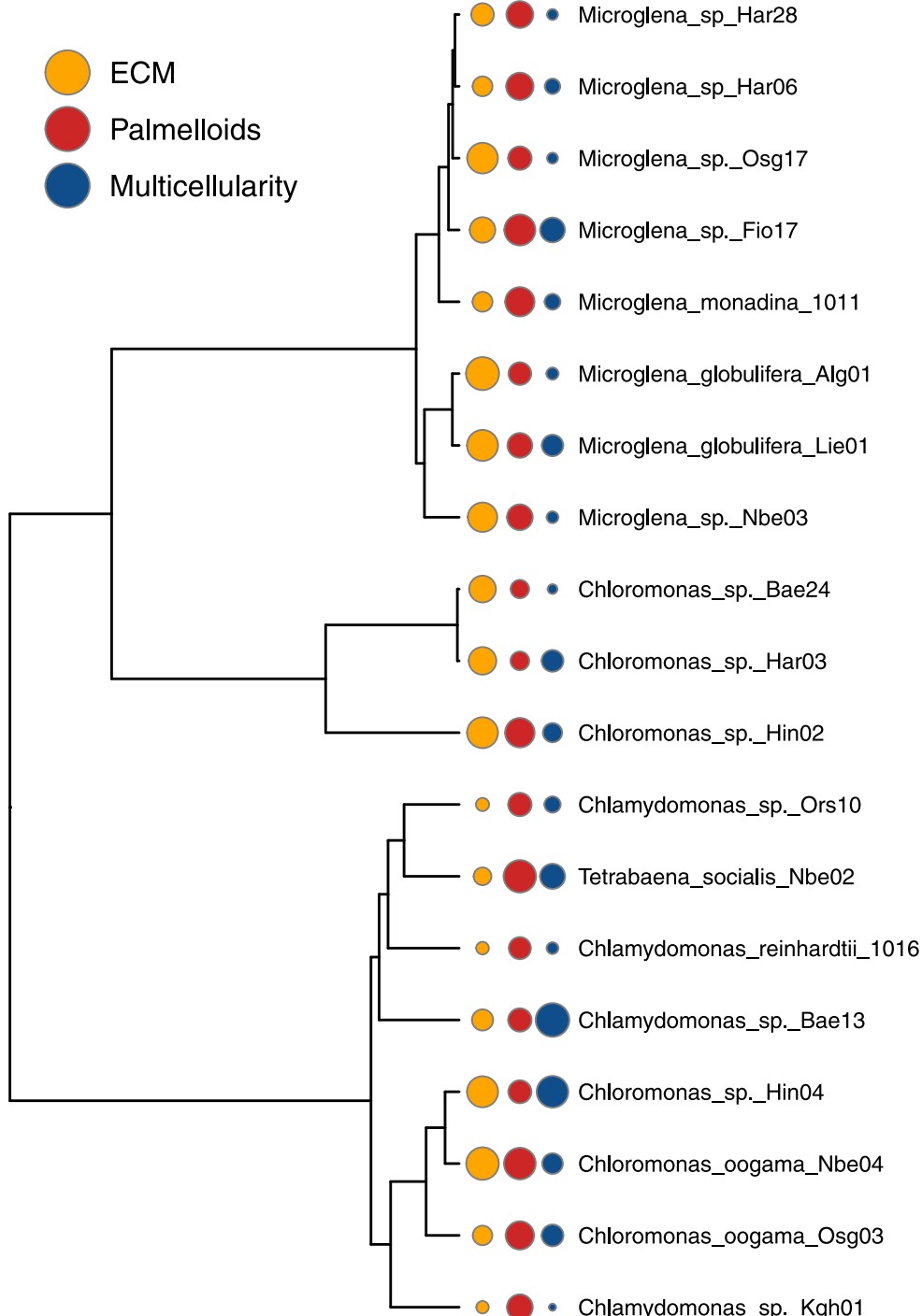

**Extended Data Fig. 1 | Phylogeny of the strains used in the experiment.** The circle sizes represent the maximum percentage of cells observed across all experimental conditions with extracellular matrix in orange, palmelloids in red and multicellular groups in blue. A maximum clade credibility tree of the 1500 trees sampled for analyses was used for plotting.

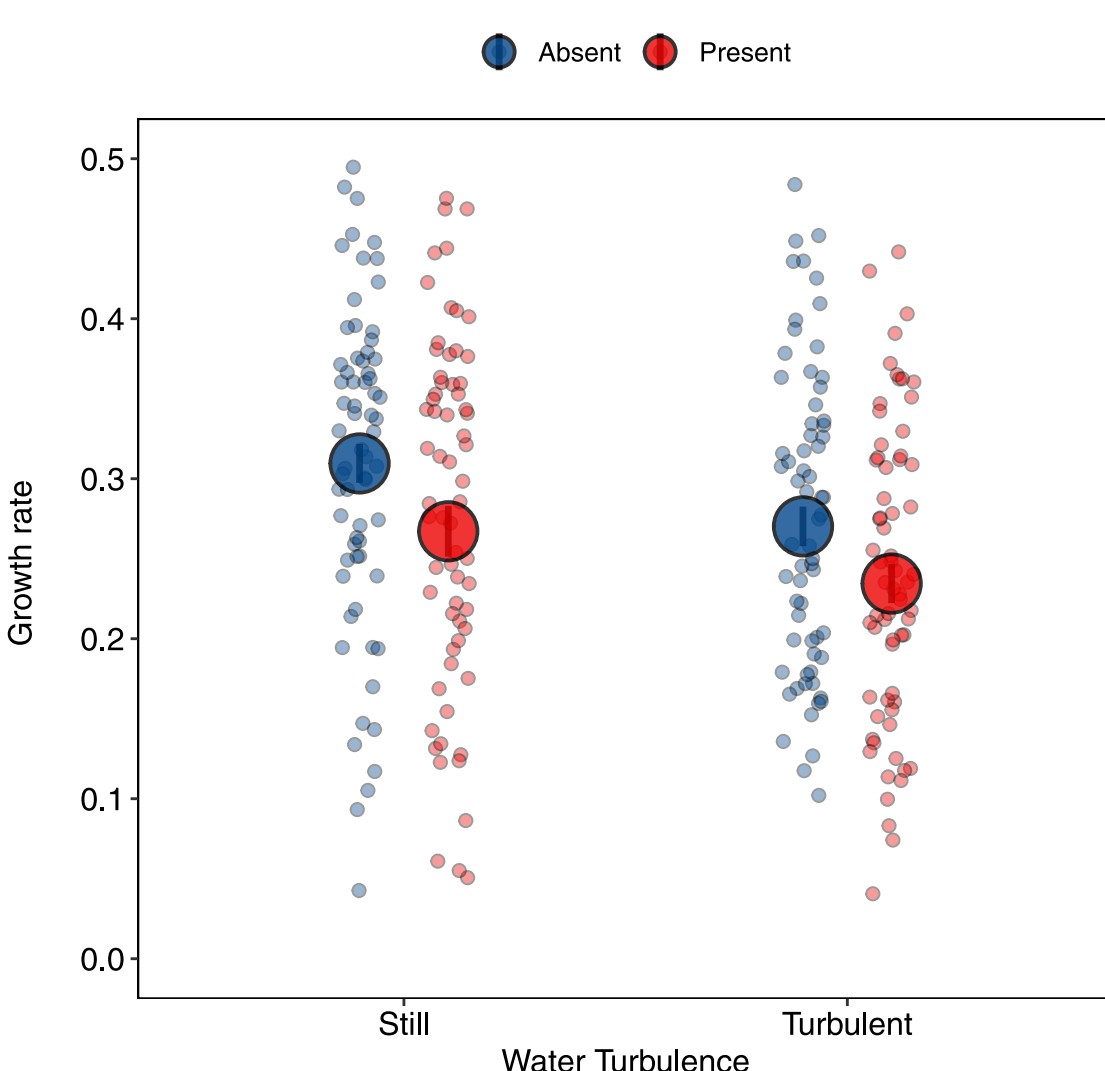

**Extended Data Fig. 2 | The reduction in growth rate caused by turbulent water conditions and predation.** Small points represent values for each of the 35 strains examined across predation, nitrogen and turbulence treatments = 274 data points. Large points with error bars (±1 SEM) are the overall treatment means.

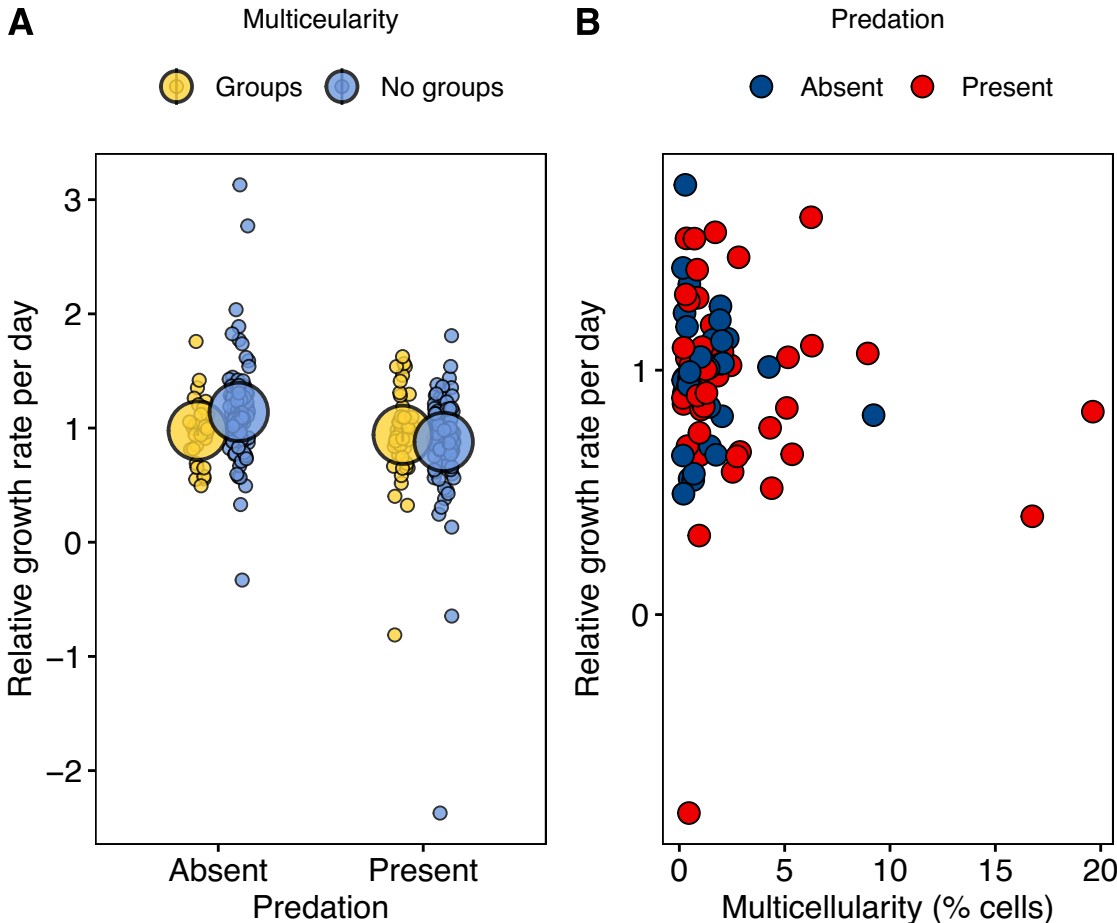

**Extended Data Fig. 3 | Multicellularity was not associated with relative growth rate under any experimental conditions. (a)** The presence of multicellular groups and (B) the percentage of cells in multicellular groups were not related to relative algal-population growth rates in the absence or presence of predators. In A, large points with error bars (±1 SEM) are the overall treatment means. Sample sizes across panels: (A) 35 strains examined across predation, nitrogen and turbulence treatments = 274 data points; (**b**) 28 strains formed groups examined across predation, nitrogen and turbulence treatments = 79 data points.

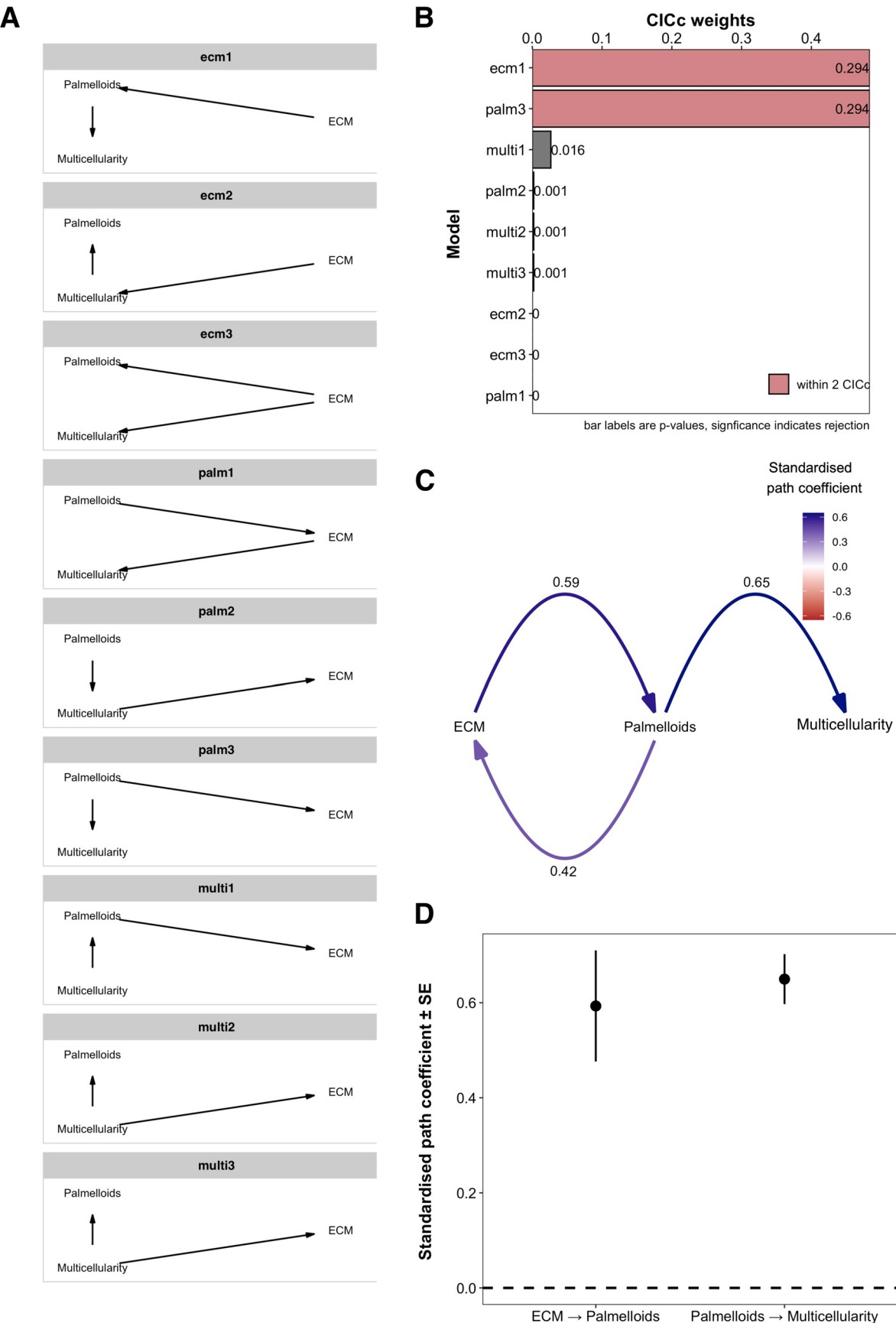

**Extended Data Fig. 4 | See next page for caption.**

**Extended Data Fig. 4 | Evaluation of the causal relationships between extracellular matrix (ECM), % cells in Palmelloids (Palmelloids) and % of cells in multicellular groups (Multicellular) using phylogenetic path analysis.** (**a**) Directed acyclic graphs (DAGs) of the different hypothetical causal models tested. (**b**) Summary of support for different causal models assessed using C-statistic information criterion corrected for small sample size (CICc) weights (higher weights = greater support). The numbers next to each bar are P values from d-separation tests examining if the model could be rejected based on the data (sample sizes: 35 strains examined across predation, nitrogen and turbulence treatments = 274 data points. P < 0.05 indicates that the model was rejected). Red bars show the top supported models that were not significantly different (a difference of less than 2 CICc units). (**c**) A path diagram of the standardised path coefficients averaged across the two best-supported models: ecm1 and palm3. (**d**) The standardised path coefficients with SEs from the best-supported model: ecm1.

# Reporting Summary

## Statistics

For all statistical analyses, confirm that the following items are present in the figure legend, table legend, main text, or Methods section.

| n/a | Confirmed | |
|---|---|---|
| ☐ | ☒ | The exact sample size (*n*) for each experimental group/condition, given as a discrete number and unit of measurement |
| ☐ | ☒ | A statement on whether measurements were taken from distinct samples or whether the same sample was measured repeatedly |
| ☐ | ☒ | The statistical test(s) used AND whether they are one- or two-sided<br>*Only common tests should be described solely by name; describe more complex techniques in the Methods section.* |
| ☐ | ☒ | A description of all covariates tested |
| ☐ | ☒ | A description of any assumptions or corrections, such as tests of normality and adjustment for multiple comparisons |
| ☐ | ☒ | A full description of the statistical parameters including central tendency (e.g. means) or other basic estimates (e.g. regression coefficient) AND variation (e.g. standard deviation) or associated estimates of uncertainty (e.g. confidence intervals) |
| ☐ | ☒ | For null hypothesis testing, the test statistic (e.g. *F*, *t*, *r*) with confidence intervals, effect sizes, degrees of freedom and *P* value noted<br>*Give P values as exact values whenever suitable.* |
| ☐ | ☒ | For Bayesian analysis, information on the choice of priors and Markov chain Monte Carlo settings |
| ☐ | ☒ | For hierarchical and complex designs, identification of the appropriate level for tests and full reporting of outcomes |
| ☐ | ☒ | Estimates of effect sizes (e.g. Cohen's *d*, Pearson's *r*), indicating how they were calculated |

*Our web collection on statistics for biologists contains articles on many of the points above.*

## Software and code

Policy information about availability of computer code

| Data collection | All data was complied using the open source software R. All versions of the R packages used together with the code and data are available at the open science framework: DOI 10.17605/OSF.IO/B9WPJ. |
|---|---|
| Data analysis | All data analysis was performed using the open source software R.. All versions of the R packages used together with the code and analysis outputs are available at the open science framework: DOI 10.17605/OSF.IO/B9WPJ. |

For manuscripts utilizing custom algorithms or software that are central to the research but not yet described in published literature, software must be made available to editors and reviewers. We strongly encourage code deposition in a community repository (e.g. GitHub). See the Nature Portfolio guidelines for submitting code & software for further information.

## Data

Policy information about availability of data

All manuscripts must include a data availability statement. This statement should provide the following information, where applicable:
- Accession codes, unique identifiers, or web links for publicly available datasets
- A description of any restrictions on data availability
- For clinical datasets or third party data, please ensure that the statement adheres to our policy

R code, data and analysis results are available at the open science framework: DOI 10.17605/OSF.IO/B9WPJ. Full citations of references in supplementary tables are given in the method references.

# Human research participants

Policy information about <u>studies involving human research participants and Sex and Gender in Research.</u>

| Reporting on sex and gender | NA |
|---|---|
| Population characteristics | NA |
| Recruitment | NA |
| Ethics oversight | NA |

Note that full information on the approval of the study protocol must also be provided in the manuscript.

# Field-specific reporting

Please select the one below that is the best fit for your research. If you are not sure, read the appropriate sections before making your selection.

☐ Life sciences      ☐ Behavioural & social sciences      ☒ Ecological, evolutionary & environmental sciences

For a reference copy of the document with all sections, see <u>nature.com/documents/nr-reporting-summary-flat.pdf</u>

# Ecological, evolutionary & environmental sciences study design

All studies must disclose on these points even when the disclosure is negative.

| Study description | We examine the environmental factors explaining the initial evolution of multicellularity in natural systems. We first experimentally examine the environmental factors that induce multicellular group formation across 35 species of unicellular green algae. Second, we test if these environmental factors predict the occurrence of obligate multicellularity across 332 species distributed across 478 lakes over the past 55-years. Specifically, we found that:<br><br>• Multicellular groups form in response to multiple environmental drivers, including nitrogen availability, water turbulence and predation.<br>• Multicellularity did not provide fitness benefits under any condition. Instead, multicellularity was associated with single cells producing extracellular matrix in high nitrogen, turbulent environments with predators that prevented the release of daughter cells.<br>• The production of extracellular matrix by single cells had a strong effect on fitness indicating that multicellular groups may arise as a by-product of selection on single cell traits.<br>• Our experimental results corresponded to nationwide patterns across Swedish lake systems. Specifically, extracellular matrix in unicellular species was related to nitrogen availability in lakes and the presence of extracellular matrix was associated with evolutionary transitions to obligate multicellularity. |
|---|---|
| Research sample | The study experimentally examined 35 species of unicellular chlorophyte algae collected from lakes in Sweden. In addition, long-term monitoring data from 478 Swedish lakes over the past 55-years of 332 species of chlorphyte alage were analysed. All data are provided in Supplementary Tables 1-4. |
| Sampling strategy | Water samples were obtained close to the shore of 20 southern Swedish lakes (Fig. 1A) using a 15 x 50 cm Apstein net with 10 μm mesh size (Hydro-Bios, Altenholz, Germany) in July and August 2016. The samples were examined for the presence of Chlamydomonas spp. at 100X and 200X in an inverted Nikon Eclipse Ts2 (Tokyo, Japan) microscope. Single swimming cells matching the general description of Chlamydomonas spp. (ca. 10-15 μm in diameter, two flagella and cup-shaped chloroplast) were isolated by micropipetting using disposable glass capillaries (Hirschmann Laborgeräte, Eberstadt, Baden-Württemberg, Germany). Cells were washed in drops of sterile-filtered lake water and placed in 100 μl 1:1 mix of WC medium (Guillard and Lorenzen 1972), modified by 0.002 mg/L Na2SeO3.5H2O (MWC + Se), and filtered lake water in 96-well culture plates (VWR, Radnor, PA, USA). Cultures were maintained at a 12:12 light:dark cycle in 20°C and 85 μmol photons m^-2^ s^-1^, transferred into larger plates as they grew, and finally placed in 25 cm2 non-treated culturing flasks (Thermo Fisher Scientific, Waltham, MA, USA) containing 30 ml MWC+Se medium.<br><br>The long-term data used is part of Sweden's national lake monitoring scheme. Details of the sampling strategies can be found at: http://www.slu.se/miljodata-MVM. |
| Data collection | The experimental data was collected by two post-docs (MSC and ML) and one research assistant (FS).<br><br>The long-term data was collected as part of Sweden's national lake monitoring scheme. Details of the sampling strategies can be found at: http://www.slu.se/miljodata-MVM. |
| Timing and spatial scale | Experimental data: The growth rates of strains and the proportion of cells in multicellular groups were measured at the start (t0), at 48 hours (t2) and at day 14-15 (t14) when the cultures reached approximate stationary phase. The experiments ran from December 2017 to February 2018. |

The long-term data involved sampling lakes across Sweden (n=478) that started 55-years ago. Full details of the number of times and when lakes were sampled is given in the supplementary materials available at the open science framework: DOI 10.17605/OSF.IO/B9WPJ.

| | |
|---|---|
| Data exclusions | No data were excluded from the analyses. |
| Reproducibility | All data analysis was performed using the open source software R. All versions of the R packages used together with fully reproducible R project scripts are available at the open science framework: DOI 10.17605/OSF.IO/B9WPJ. |
| Randomization | Our experimental designed selected strains of algae that had different geographic and phylogenetic history. Within these restrictions we randomly choose strains to study. |
| Blinding | Experimenters were blind to the treatments when collecting data. In addition, phenotypic measurements were automated using machines (e.g. FlowCam). |

Did the study involve field work? ☐ Yes ☒ No

# Reporting for specific materials, systems and methods

We require information from authors about some types of materials, experimental systems and methods used in many studies. Here, indicate whether each material, system or method listed is relevant to your study. If you are not sure if a list item applies to your research, read the appropriate section before selecting a response.

## Materials & experimental systems

| n/a | Involved in the study |
|---|---|
| ☒ | Antibodies |
| ☒ | Eukaryotic cell lines |
| ☒ | Palaeontology and archaeology |
| ☐ | ☒ Animals and other organisms |
| ☒ | Clinical data |
| ☒ | Dual use research of concern |

## Methods

| n/a | Involved in the study |
|---|---|
| ☒ | ChIP-seq |
| ☒ | Flow cytometry |
| ☒ | MRI-based neuroimaging |

## Animals and other research organisms

Policy information about studies involving animals; ARRIVE guidelines recommended for reporting animal research, and Sex and Gender in Research

| | |
|---|---|
| Laboratory animals | NA. The study is on algae. |
| Wild animals | NA. The study is on algae. |
| Reporting on sex | NA. |
| Field-collected samples | Yes. Chlorophyte algae were collected from natural lakes. |
| Ethics oversight | NA |

Note that full information on the approval of the study protocol must also be provided in the manuscript.

