## [Peer Review File · Nature Ecology & Evolution]

Peer Review Information

Journal: Nature Ecology & Evolution

Manuscript Title: Single cell adaptations shape evolutionary transitions to multicellularity in green algae

Corresponding author name(s):

Editorial Notes:

Reviewer Comments & Decisions:

Decision Letter, initial version:

12th December 2022

Dear Charlie,

Your manuscript entitled "Single cell adaptations shape evolutionary transitions to multicellularity" has now been seen by 3 reviewers, whose comments are attached. The reviewers have raised a number of concerns which will need to be addressed before we can offer publication in Nature Ecology & Evolution. We will therefore need to see your responses to the criticisms raised and to some editorial concerns, along with a revised manuscript, before we can reach a final decision regarding publication.

We therefore invite you to revise your manuscript taking into account all reviewer and editor comments. Please highlight all changes in the manuscript text file in Microsoft Word format.

* If you have not done so already please begin to revise your manuscript so that it conforms to our Article format instructions at <http://www.nature.com/natecolevol/info/final-submission>. Refer also to any guidelines provided in this letter.

[REDACTED]

2Note: This URL links to your confidential home page and associated information about manuscripts you may have submitted, or that you are reviewing for us. If you wish to forward this email to co-authors, please delete the link to your homepage.

Nature Ecology & Evolution is committed to improving transparency in authorship. As part of our efforts in this direction, we are now requesting that all authors identified as 'corresponding author' on published papers create and link their Open Researcher and Contributor Identifier (ORCID) with their account on the Manuscript Tracking System (MTS), prior to acceptance. ORCID helps the scientific community achieve unambiguous attribution of all scholarly contributions. You can create and link your ORCID from the home page of the MTS by clicking on 'Modify my Springer Nature account'. For more information please visit www.springernature.com/orcid.

[REDACTED]

Reviewer expertise:

Reviewer #1: algal evolution and development

Reviewer #2: evolution of multicellularity, experimental

Reviewer #3: evolution of multicellularity, genomics

Reviewers' comments:

Reviewer #1 (Remarks to the Author):

The emergence of multicellularity is one of the most spectacular of the major transitions during the evolutionary history of life. The paper aims to understand what are the ecological advantages to the initial formation of multicellular groups, and why multicellular groups occur in some lineages, but not others. These are important outstanding questions in multicellularity research that are of interest to a broad audience interested in both the experimental and theoretical aspects of this field of research.

2The authors used green algae, and phylogenetic comparative analyses to examine the ecological conditions that explain multicellular group formation using a combination of experiments and long-term lake monitoring data. Specifically, they test which environmental conditions promote multicellular group formation and the extent to which this was explained by past ecological and evolutionary history. They investigate the fitness of multicellularity under different environmental conditions and whether multicellular group formation was explained by changes in cellular morphology in response to environmental conditions. Finally, they used long-term database (55-years) of several hundred lakes to validate the environmental conditions influencing the evolution of multicellularity across green algae.

I appreciate the large-scale effort to characterize the transition from uni- to (simple) multicellularity. The dataset is rather impressive. However, the paper needs substantial improvement in the clarification of the experiments, and I am not convinced it provides a sufficiently important advance to the community. It is disappointing that the paper is rather descriptive, and the conclusions are based on (admittedly highly replicated/large datasets) correlations rather than causality. Unless I missed it, I could see no effort to proceed on detailed analysis, for example on the cell biology of the group formation, and the links between ECM, group formation and palmelloids are vague. As it is, the paper falls short in shedding light into the proximate or ultimate mechanisms underlying multicellular group formation, and I end up with more questions than answers.

I could not understand the link between palmelloids and multicellular groups, for example the following sentence is confusing: "(...) Instead, fitness was related to the production of extracellular matrix (ECM) that resulted in cells being retained within their mother cell wall (palmelloids) and subsequently the formation of larger multicellular groups." Do you mean the palmelloids precede the multicellular group formation? it would be important to understand how do the palmelloids originate – from meiosis? What is the difference with "multicellular groups"? Is this a phase of the life cycle of these algae? Are palmelloids composed always of 4 cells?

The main conclusions in several sections in the paper are very vague, for example: "Together, these results suggest that the propensity of different strains to form multicellular groups is evolutionary conserved across species, but the degree to which multicellularity is expressed is determined by the interaction between multiple environmental factors." I am afraid I learn very little...

Line 139: how do you distinguish between growth rate and "predation", i.e., how do you know that reduced number of cells are not the result of predation (you assume that the fitness is reduced due to decreased growth rate, but it could be that the growth rate is higher but many cells are being predated?)

Where are the results showing that there is higher fitness in palmelloids and where is the evidence that ECM is the driver of the increased fitness? This is very speculative. Also, I don't understand the term "subsequently" – do you mean first the cells produce palmelloids and then produce multicellular groups?

"Multicellular group formation was widespread among unicellular algae, occurring in 91% of the strains." - where is the data supporting this statement? In supplemental table?

Clarifications are needed in the way the paper is organized. Last paragraph of introduction explains how the paper is organized but this is uncoupled with the results section. It is weird to have reference to Tables in the end of introduction.

In the first sentence in the results section, it is unclear what dataset was used. Is this related to the test "we tested which environmental conditions promote multicellular group formation and the extent to which this was explained by past ecological and evolutionary history"?

Supplementary Table 4 (and elsewhere): when you mention multicellular groups you mean more than four cells (so not palmelloids)?

Figure 2 - How do the authors explain the bimodal distribution of predator growth rate?

"We also examined if strains with a higher prevalence of multicellular groups reduced the growth of predators." – what are the premises to expect this???

The authors mention obligate multicellularity in the abstract and introduction. Line 34 "ECM production across unicellular species was related to nitrogen availability, which in turn was associated with evolution of obligate multicellularity." but I don't see any result related to "obligate multicellularity" and which species are facultative or obligate multicellular in your experiment? – did I miss it? The link between ECM, palmelloid persistence and nitrogen availability put forward by the paper is based on correlations, no causal links are presented unless I missed it.

Minor comments

Extended figure 3 – typo in title

Line 118 "However, turbulence and nitrate levels both had a greater effect on multicellularity than predation" why not show this in a figure? (reference only to a supplemental table)

Legends of supplemental tables are incomplete. For example what does "SGR" mean in Table S1?

Typo in legend of Table S1.

Abstract: 'palleloid' is a typo? You mean palmelloid?

Reviewer #2 (Remarks to the Author):

The manuscript seeks to experimentally identify environmental factors that influence the formation of multicellular algae and to test whether predictions derived from these experiments bear out in the distributions of unicellular and multicellular algae in nature. The writing is clear, the experiments rigorous as best I can tell, and the results interpreted with appropriate caution. The manuscript identifies a real knowledge gap and goes a substantial distance toward filling it. It is true that we know surprisingly little about the environmental factors that induce multicellularity, and this is a step in the right direction. I'm kind of blown away by the experimental design, which included eight sets of environmental conditions for an impressive number of genetically distinct strains. The comparisons with natural distributions is by necessity correlative, but the results are intriguing and not over-interpreted.

Throughout the manuscript, I feel that the distinction between plastic and evolved changes could have been clearer. I assume in a 14-day experiment we're mainly talking about plastic changes (and I think this is consistent with the Discussion around line 285), but in a manuscript that addresses both plastic and evolved changes, it would be good to be completely clear about which is which.

The supplementary tables need much more explanation. These look like raw output of statistics software, not publication-ready tables. Statistical tests are not explained in the captions, headings are not defined, and abbreviations are not spelled out. I assume R^2 is R^2 , but I don't know what DIC is, or I^2 %, or posterior mode. In some cases it's not 100% clear what the response variable is.

Related to this, I found the statistics reported in the manuscript opaque. I don't know what a posterior mode of 32.3 means. Is this the percent of variation explained by that factor, as lines 114-115 and 203-204 seem to suggest? But then some values are negative, and some over 100, so that can't be right. A few lines explaining what these statistics mean would make the results easier to understand.

4I would suggest moving some of the information in the Methods earlier in the paper. For example, the Results and Discussion would be easier to understand if the experimental design were explained earlier (an artifact of the Methods-at-the-end format).

Reviewer #3 (Remarks to the Author):

Review of "Single cell adaptations shape evolutionary transitions to multicellularity" by Cornwallis et al.

The authors provide a tour-de-force analysis of potential environmental factors inducing multicellularity in wide range (35!) of unicellular green algae. The experiments are well done, the results are clear and the interpretations are also adequate. I only have very minor comments:

1. Title. Because the work is specific to this taxonomic group, I propose "Single cell adaptations shape evolutionary transitions to multicellularity in green algae" or something similar.
2. Line 151. "The apparent lack of benefits of multicellularity raises the question". I am not sure what they mean here. From this study? From other studies?
3. Line 153. "studies have highlighted that the evolution of multicellularity involved the co-option of genes regulating cell replication". I think this has been shown in green algae. This should be clarified..
4. Line 237. Also specify that "formation of multicellular groups" in green algae. The authors can hypothesize or discuss whether this may also be relevant to other multicellular acquisitions, but their work is limited to green algae.
5. Line 270. "Bacterial symbioses..." The work cited is about "bacterial interactions". I propose using that instead of symbiosis.

*****END*****

Author Rebuttal to Initial comments

Response to reviewers

We are very grateful to the editors and reviewers for their positive and constructive feedback. Their input has been extremely valuable, and we feel the paper is substantially improved. Below are the details of the specific changes we have made to address their comments. Line numbers in word documents with track

5changes are sometimes difficult to reproduce across computers. Therefore, all line numbers refer to the pdf version of the revised manuscript.

Reviewer #1 (Remarks to the Author):

The emergence of multicellularity is one of the most spectacular of the major transitions during the evolutionary history of life. The paper aims to understand what are the ecological advantages to the initial formation of multicellular groups, and why multicellular groups occur in some lineages, but not others. These are important outstanding questions in multicellularity research that are of interest to a broad audience interested in both the experimental and theoretical aspects of this field of research.

The authors used green algae, and phylogenetic comparative analyses to examine the ecological conditions that explain multicellular group formation using a combination of experiments and long-term lake monitoring data. Specifically, they test which environmental conditions promote multicellular group formation and the extent to which this was explained by past ecological and evolutionary history. They investigate the fitness of multicellularity under different environmental conditions and whether multicellular group formation was explained by changes in cellular morphology in response to environmental conditions. Finally, they used long-term database (55-years) of several hundred lakes to validate the environmental conditions influencing the evolution of multicellularity across green algae.

We are very thankful to the referee for their supportive words and pleased they find our work is of general importance and interest to a broad audience.

I appreciate the large-scale effort to characterize the transition from uni- to (simple) multicellularity. The dataset is rather impressive. However, the paper needs substantial improvement in the clarification of the experiments, and I am not convinced it provides a sufficiently important advance to the community. It is disappointing that the paper is rather descriptive, and the conclusions are based on (admittedly highly replicated/large datasets) correlations rather than causality. Unless I missed it, I could see no effort to proceed on detailed analysis, for example on the cell biology of the group formation, and the links between ECM, group formation and pameloids are vague. As it is, the paper falls short in shedding light into the proximate or ultimate mechanisms underlying multicellular group formation, and I end up with more questions than answers.

We appreciate the constructive feedback and are grateful for the opportunity to clarify the novel insights provided by our work. To address this issue, we have revised the text in the introduction (lines 77-96) and included a new paragraph in the discussion (398-418), that clarifies how our results advance the field. To briefly summarise, our study makes the following contributions:

1) Previous research has typically focused on the effect of single environmental factors on multicellularity. This has meant that little is known about the relative importance of different environmental factors and how their interactions alter selection for multicellularity. Our study helps fill this gap, providing insight into the effect of interactions between predation, nutrients and environmental turbulence on ECM production, retention of cells within palmelloids, and the formation of multicellular groups.

2) Our study is the first to conduct experiments across a wide range of species, which was crucial to characterising the relationships ECM, palmelloid persistence and multicellularity. Previous experimental research has typically focused on single species and therefore has not been able to resolve how variation in ECM and palmelloid persistence among unicellular species relate to facultatively multicellular responses.

3) Recent prominent reviews (see forward by Andrew Knoll in Herron *et. al.* 2022¹) have highlighted the need for studies examining the natural environments associated with the evolution of multicellularity. Our analyses of the distribution of unicellular and multicellular species across national scales provides such insight. While this dataset is, by definition, correlative, integrating such analyses with experimental evidence offers an important step forward in our understanding of how the environment shapes the evolution of single-cell traits and their link to multicellularity.

4) ECM, palmelloid formation and multicellularity were not experimentally manipulated and therefore the causality underlying these relationships remains to be further investigated. Prior to our study, it was unclear how these traits were related to each other and how they varied across species. It would therefore have been difficult to choose the relevant species and conditions to investigate the causality underlying the associations between ECM, palmelloids and multicellular group formation. However, our paper provides a robust foundation for further experimental work and we hope that it will stimulate more research in that direction. We have clarified this point in lines 416-418.

In addition, we have added extra analyses to gain more insight into the causal relationships between ECM, palmelloid persistence and multicellularity. Specifically, we used phylogenetic path analysis that enables different hypothetical causal models to be evaluated. Path analysis starts by defining the potential competing causal relationships ('path diagrams') between candidate sets of variables (e.g. ECM affects rates of palmelloid formation, and palmelloid formation affects rates of multicellularity. See extended data figure 4 for the causal models investigated). Evidence for these models is evaluated using model fit statistics from a series of regressions that represent the different causal relationships. In line with our previous interpretation of the results, this analysis showed that best supported causal model is one where ECM affects palmelloid formation and palmelloid formation affects multicellularity. Details of these analyses are presented in the results (lines 292-300), methods (lines 1139-1159) and Extended data figure 4.

I could not understand the link between palmelloids and multicellular groups, for example the following sentence is confusing: “(...) Instead, fitness was related to the production of extracellular matrix (ECM) that resulted in cells being retained within their mother cell wall (palmelloids) and subsequently the formation of larger multicellular groups.” Do you mean the palmelloids precede the multicellular group formation? it would be important to understand how do the palmelloids originate – from meiosis? What is the difference with “multicellular groups”? Is this a phase of the life cycle of these algae? Are palmelloids composed always of 4 cells?

This is a key point and we are grateful to the referee for pointing out this needed clarification. Palmelloids form by a mother cell going through two rounds of mitosis to produce four daughter cells. It is a normal stage of the unicellular lifecycle and has been observed in all unicellular chlorophyte species^{2,3}. Therefore, we defined palmelloids as 2-4 cells and multicellular groups as more than four cells, which are not part of the typical unicellular lifecycle. In accordance with the reviewer's suggestion, we have clarified how palmelloids form and how we define palmelloids versus multicellular groups in the introduction (lines 30-32, 116-121).

The main conclusions in several sections in the paper are very vague, for example: “Together, these results suggest that the propensity of different strains to form multicellular groups is evolutionary conserved across species, but the degree to which multicellularity is expressed is determined by the interaction between multiple environmental factors.” I am afraid I learn very little...

We have now revised the main conclusion of this section to clarify the take-home message (lines 169-172).

Line 139: how do you distinguish between growth rate and “predation”, i.e., how do you know that reduced number of cells are not the result of predation (you assume that the fitness is reduced due to decreased growth rate, but it could be that the growth rate is higher but many cells are being predated?)

As the referee points out, changes in the number of cells may occur due to predators consuming cells, or due to strains adjusting their growth rate. If multicellularity is advantageous, then it is predicted that strains that form multicellular groups will have higher cell numbers at the end of the experiment regardless of the mechanism: growth rate adjustment versus protection from predators.

We did not find that cell numbers were related to rates of multicellularity, which goes against the predominant view that multicellular groups provide a fitness advantage when predators are present. We realise that using the term growth rate made it ambiguous whether we were referring to ‘population’ growth rate or the growth rate of individual cells. Therefore, we have revised the text throughout to clarify that we are concerned with changes in population growth rather than the growth of individual cells (e.g. lines 174-229).

Where are the results showing that there is higher fitness in palmelloids and where is the evidence that ECM is the driver of the increased fitness? This is very speculative. Also, I don’t understand the term “subsequently” – do you mean first the cells produce palmelloids and then produce multicellular groups?

We are grateful to the referee for pointing out that this section was unclear. We have revised the text to clarify how palmelloids form and that it precedes the formation of multicellular groups (lines 29-33, 234-251).

The results showing the fitness effects of ECM and palmelloid formation can be found in the results section entitled “The fitness payoffs of ECM and palmelloid formation” (lines 301-337 revised version) and in figure 4.

“Multicellular group formation was widespread among unicellular algae, occurring in 91% of the strains.” - where is the data supporting this statement? In supplemental table?

9Clarifications are needed in the way the paper is organized. Last paragraph of introduction explains how the paper is organized but this is uncoupled with the results section. It is weird to have reference to Tables in the end of introduction.

We have now added a reference to Supplementary tables 1 and 2 that provide details of the 32 experimental strains (91%) that formed multicellular groups and under which conditions. We have also checked and amended the last paragraph of the introduction that outlines what we do and adjusted the structure of the results section to match (lines 149-172).

We refer to supplementary tables 1 to 4 at the end of the introduction, which contain the raw data collected to test each aim. We placed these references here as we thought readers may want to inspect the data prior to looking at the results, for example, if they are interested in which algae species we studied. Hence, we feel it is clearest to cite these tables when we first mention the data, but are happy to move these references to the start of the results section if this better fits editorial style.

In the first sentence in the results section, it is unclear what dataset was used. Is this related to the test “we tested which environmental conditions promote multicellular group formation and the extent to which this was explained by past ecological and evolutionary history”?

Our intention was to first provide evidence of how widespread multicellular group formation was across the studied algae species. We are grateful to the referee for pointing out that this was confusing and not closely connected enough to our first aim. To rectify this issue, we have restructured the paragraph and revised the text (lines 149-172).

Supplementary Table 4 (and elsewhere): when you mention multicellular groups you mean more than four cells (so not palmelloids)?

This is correct. We have made this more explicit earlier on in the manuscript, explaining how palmelloids are part of the unicellular lifecycle and different from multicellular groups (lines 31-33, 116-121. See also response to comment above).

Figure 2 - How do the authors explain the bimodal distribution of predator growth rate?

10“We also examined if strains with a higher prevalence of multicellular groups reduced the growth of predators.” – what are the premises to expect this??

The bimodal distribution of predator growth rates is explained by *Brachinous* being unable to survive on some *Microglena* strains (figure 2 legend “In D, predators were unable to survive on some strains of the *Microglena* clade, generating a bimodal distribution of predator growth rates in the no multicellular group category. We suspect this is either because they produce toxins or provide limited nutrients for *Brachinous*”). We have some follow-up pilot data that suggests this is because these algae strains produce toxins, but it is currently too preliminary to include.

Our premise for examining predator numbers was that if multicellularity is a defence against predation, by preventing ingestion or digestion, then predators should starve. We realise that using the term growth rate was confusing and have changed this to ‘population’ growth of predators throughout. In addition, we have clarified our rationale behind analysing predation population growth rates (lines 224-225).

The authors mention obligate multicellularity in the abstract and introduction. Line 34 “ECM production across unicellular species was related to nitrogen availability, which in turn was associated with evolution of obligate multicellularity.” but I don’t see any result related to “obligate multicellularity” and which species are facultative or obligate multicellular in your experiment? – did I missed it?

The strains used in experiments were all unicellular when collected from lakes. Consequently, our experiments studied unicellular species and their facultative multicellular responses. In contrast, our analyses of the long-term lake monitoring data examined species described in the literature as unicellular and obligately multicellular.

We have now clarified that our experiments examined unicellular species and their facultatively multicellular responses and that the analyses of the 332 species of algae found in lakes examine the differences between unicellular and obligately multicellular species (lines 136-142).

The link between ECM, palmiloid persistence and nitrogen availability put forward by the paper is based on correlations, no causal links are presented unless I missed it.

This is an important point and we are grateful to the referee for highlighting that we need to be clearer about which of our results provide causal versus correlative insight (see also response to comment above).

The referee is correct that our analyses of the long-term data are correlative. The effort required to collect this data (332 species, 478 lakes over 55 years) was enormous and has allowed general features of lake ecosystems to be described in detail (here and elsewhere). Including extra approaches to understand the causal structure of results, such as experiments, was unfortunately not feasible at this scale. Therefore, we decided to integrate such analyses with smaller-scale experiments that allow some causal inference.

By manipulating environmental variables, our experiments show that nitrogen availability has a causal effect on palmelloid persistence and multicellular group formation. In addition, an important advance provided by our work is that experiments were conducted across species that varied with respect to the presence and absence of ECM. Analysing the responses of species with and without ECM, and showing that ECM around single cells influences the retention of cells within palmelloids, offers novel evidence of the mechanisms linking ECM with multicellularity. That said, it was beyond the scope of our study to directly manipulate ECM production or the persistence of palmelloids, which indeed deserves to be investigated more extensively in future studies.

To address this point within the scope of the current manuscript we have added extra analyses (phylogenetic path analysis) that examine the causal structure of our data (lines 292-300, 1139-1159 and Extended data figure 4). Characterising how these traits are related to each other across species, which was not previously known, is an important precursor to conducting more detailed experimental work. Manipulating ECM and palmelloid retention is potentially feasible, for example through gene knockouts and knockins, but this is challenging to do in multiple species. We hope our paper will provide the foundations for such studies enabling appropriate focal species to be chosen. We have clarified in the discussion what causal versus correlative evidence we provide for the relationship between ECM, palmelloid persistence, nitrogen availability and multicellularity, and how we hope this will facilitate future work (lines 398-418).

Minor comments

Extended figure 3 – typo in title

12Addressed.

Line 118 “However, turbulence and nitrate levels both had a greater effect on multicellularity than predation” why not show this in a figure? (reference only to a supplemental table)

Thanks for the suggestion. We now reference figure 1 that shows this.

Legends of supplemental tables are incomplete. For example what does “SGR” mean in Table S1?

We have now revised all Supplementary table legends that give more explanation to their content and give full details of acronyms used.

Typo in legend of Table S1.

Addressed.

Abstract: ‘palleloid’ is a typo? You mean palmelloid?

Addressed.

Reviewer #2 (Remarks to the Author):

The manuscript seeks to experimentally identify environmental factors that influence the formation of multicellular algae and to test whether predictions derived from these experiments bear out in the distributions of unicellular and multicellular algae in nature. The writing is clear, the experiments rigorous as best I can tell, and the results interpreted with appropriate caution. The manuscript identifies a real knowledge gap and goes a substantial distance toward filling it. It is true that we know surprisingly little about the environmental factors that induce multicellularity, and this is a step in the right direction. I'm kind of blown away by the experimental design, which included eight sets of environmental conditions for an impressive number of genetically distinct

13strains. The comparisons with natural distributions is by necessity correlative, but the results are intriguing and not over-interpreted.

We very much appreciate the kind words.

Throughout the manuscript, I feel that the distinction between plastic and evolved changes could have been clearer. I assume in a 14-day experiment we're mainly talking about plastic changes (and I think this is consistent with the Discussion around line 285), but in a manuscript that addresses both plastic and evolved changes, it would be good to be completely clear about which is which.

This is an important point, and we are grateful to the referee for pointing out that this required clarification. We have now edited the text in several places to distinguish between plastic and evolved responses (lines 136-142 and 389-395).

The supplementary tables need much more explanation. These look like raw output of statistics software, not publication-ready tables. Statistical tests are not explained in the captions, headings are not defined, and abbreviations are not spelled out. I assume R^2 is R^2 , but I don't know what DIC is, or I^2 %, or posterior mode. In some cases it's not 100% clear what the response variable is.

We have now completely revised the headings of the supplementary tables to provide more explanation and definitions of column headings.

Related to this, I found the statistics reported in the manuscript opaque. I don't know what a posterior mode of 32.3 means. Is this the percent of variation explained by that factor, as lines 114-115 and 203-204 seem to suggest? But then some values are negative, and some over 100, so that can't be right. A few lines explaining what these statistics mean would make the results easier to understand.

We apologise that this was not clearer. Posterior modes are a way of summarizing the posterior distribution of parameter estimates that are produced by Bayesian analyses. Therefore, in some cases they can be regression coefficients (positive or negative), or if they refer to random effects, they are variance

components. We have now edited the text throughout to make it more explicit what statistics refer to (e.g. lines 138-142, 154-156, 166-167).

I would suggest moving some of the information in the Methods earlier in the paper. For example, the Results and Discussion would be easier to understand if the experimental design were explained earlier (an artifact of the Methods-at-the-end format).

Many thanks for the suggestion. We have now expanded the description of our experimental design at the end of introduction.

Reviewer #3 (Remarks to the Author):

Review of “Single cell adaptations shape evolutionary transitions to multicellularity” by Cornwallis et al.

The authors provide a tour-de-force analysis of potential environmental factors inducing multicellularity in wide range (35!) of unicellular green algae. The experiments are well done, the results are clear and the interpretations are also adequate. I only have very minor comments:

Many thanks for the very supportive feedback!

1. Title. Because the work is specific to this taxonomic group, I propose "Single cell adaptations shape evolutionary transitions to multicellularity in green algae" or something similar.

Addressed.

2. Line 151. “The apparent lack of benefits of multicellularity raises the question”. I am not sure what they mean here. From this study? From other studies?

We have now clarified that we mean our experiments (line 231).

3. Line 153. “studies have highlighted that the evolution of multicellularity involved the co-option of genes regulating cell replication”. I think this has been shown in green algae. This should be clarified..

15Addressed.

4. Line 237. Also specify that “formation of multicellular groups” in green algae. The authors can hypothesize or discuss whether this may also be relevant to other multicellular acquisitions, but their work is limited to green algae.

Addressed.

5. Line 270. “Bacterial symbioses...” The work cited is about “bacterial interactions”. I propose using that instead of symbiosis.

Addressed.

References

1. Herron, M. D., Conlin, P. L. & Ratcliff, W. C. *The Evolution of Multicellularity*. (Routledge, 2022).
2. Tikkanen, T. & Willén, T. *Växtplanktonflora*. (Statens naturvårdsverk, 1992).
3. Kirchner. *Algen. In: Kryptogamen-Flora von Schlesien. Part 1. (Cohn, F. Eds)*. vol. 2 (Verlag J. U. Kern, Breslau, 1878).

Decision Letter, first revision:

15th February 2023

Dear Charlie,

Thank you for submitting your revised manuscript "Single cell adaptations shape evolutionary transitions to multicellularity in green algae" (NATECOLEVOL-221017795A). It has now been seen

16again by the original reviewers and their comments are below. The reviewers find that the paper has improved in revision, and therefore we'll be happy in principle to publish it in Nature Ecology & Evolution, pending minor revisions to satisfy the reviewers' final requests and to comply with our editorial and formatting guidelines.

[REDACTED]

Reviewer #1 (Remarks to the Author):

I appreciate the authors' efforts to clarify the points raised in my previous review. I have reviewed the revised manuscript as well as the authors' attempts to address my and the other reviewers suggestions. I find the expanded explanations in the text helpful in clarifying many of the points that were initially unclear.

Some minor comments:

line 217) 'To investigate this, we conducted a second experiment on the same 35 strains were exposed to varying numbers

of predators (control = 0 per mL, low = ca. 15 individuals mL⁻¹ and high = ca. 50219 individuals mL⁻¹). ' (sentence reads weirdly)

line 248) "mother cells divide into four daughter cells " consider adding "mitotically" to avoid confusion with meiotic cell division

Reviewer #2 (Remarks to the Author):

The authors have addressed all of the issues I identified to my satisfaction. These changes and those made in response to the other two reviewers have considerably improved the manuscript. Although I agree with Reviewer 1 that the work is largely correlative and that it leaves many questions unanswered, this does not in my mind keep it from being a valuable advance. I would like to see this information made available to the scientific community, and I am content to leave more rigorous tests of causation and mechanism to future work by these and/or other researchers.

Reviewer #3 (Remarks to the Author):

I am happy with how the authors addressed all the reviewer's concerns.

Our ref: NATECOLEVOL-221017795A

22nd February 2023

Dear Dr. Cornwallis,

Thank you for your patience as we've prepared the guidelines for final submission of your Nature Ecology & Evolution manuscript, "Single cell adaptations shape evolutionary transitions to multicellularity in green algae" (NATECOLEVOL-221017795A). Please carefully follow the step-by-step instructions provided in the attached file, and add a response in each row of the table to indicate the changes that you have made. Please also check and comment on any additional marked-up edits we have proposed within the text. Ensuring that each point is addressed will help to ensure that your revised manuscript can be swiftly handed over to our production team.

****We would like to start working on your revised paper, with all of the requested files and forms, as soon as possible (preferably within two weeks). Please get in contact with us immediately if you anticipate it taking more than two weeks to submit these revised files.****

In recognition of the time and expertise our reviewers provide to Nature Ecology & Evolution's editorial process, we would like to formally acknowledge their contribution to the external peer review of your manuscript entitled "Single cell adaptations shape evolutionary transitions to multicellularity in green algae". For those reviewers who give their assent, we will be publishing their names alongside the published article.

Nature Ecology & Evolution offers a Transparent Peer Review option for new original research manuscripts submitted after December 1st, 2019. As part of this initiative, we encourage our authors to support increased transparency into the peer review process by agreeing to have the reviewer comments, author rebuttal letters, and editorial decision letters published as a Supplementary item. When you submit your final files please clearly state in your cover letter whether or not you would like to participate in this initiative. Please note that failure to state your preference will result in delays in accepting your manuscript for publication.

18Cover suggestions

As you prepare your final files we encourage you to consider whether you have any images or illustrations that may be appropriate for use on the cover of Nature Ecology & Evolution.

Nature Ecology & Evolution has now transitioned to a unified Rights Collection system which will allow our Author Services team to quickly and easily collect the rights and permissions required to publish your work. Approximately 10 days after your paper is formally accepted, you will receive an email in providing you with a link to complete the grant of rights. If your paper is eligible for Open Access, our Author Services team will also be in touch regarding any additional information that may be required to arrange payment for your article.

Please note that *Nature Ecology & Evolution* is a Transformative Journal (TJ). Authors may publish their research with us through the traditional subscription access route or make their paper immediately open access through payment of an article-processing charge (APC). Authors will not be required to make a final decision about access to their article until it has been accepted. [Find out more about Transformative Journals](https://www.springernature.com/gp/open-research/transformative-journals)

Authors may need to take specific actions to achieve [compliance with funder and institutional open access mandates](https://www.springernature.com/gp/open-research/funding/policy-compliance-faqs). If your research is supported by a funder that requires immediate open access (e.g. according to [Plan S principles](https://www.springernature.com/gp/open-research/plan-s-compliance)) then you should select the gold OA route, and we will direct you to the compliant route where possible. For authors selecting the subscription publication route, the journal's standard licensing terms will need to be accepted, including [self-archiving-and-license-to-publish](https://www.nature.com/nature-portfolio/editorial-policies/self-archiving-and-license-to-publish). Those licensing terms will supersede any other terms that the author or any third party may assert apply to any version of the manuscript.

For information regarding our different publishing models please see our <https://www.springernature.com/gp/open-research/transformational-journals> Transformative Journals page. If you have any questions about costs, Open Access requirements, or our legal forms, please contact ASJournals@springernature.com.

[REDACTED]

[REDACTED]

Reviewer #1:

Remarks to the Author:

I appreciate the authors' efforts to clarify the points raised in my previous review. I have reviewed the revised manuscript as well as the authors' attempts to address my and the other reviewers suggestions. I find the expanded explanations in the text helpful in clarifying many of the points that were initially unclear.

Some minor comments:

line 217) 'To investigate this, we conducted a second experiment on the same 35 strains were exposed to varying numbers of predators (control = 0 per mL, low = ca. 15 individuals mL⁻¹ and high = ca. 50219 individuals mL⁻¹). ' (sentence reads weirdly)

line 248) "mother cells divide into four daughter cells " consider adding "mitotically" to avoid confusion with meiotic cell division

Reviewer #2:

Remarks to the Author:

The authors have addressed all of the issues I identified to my satisfaction. These changes and those made in response to the other two reviewers have considerably improved the manuscript. Although I agree with Reviewer 1 that the work is largely correlative and that it leaves many questions unanswered, this does not in my mind keep it from being a valuable advance. I would like to see this information made available to the scientific community, and I am content to leave more rigorous tests of causation and mechanism to future work by these and/or other researchers.

20Reviewer #3:

Remarks to the Author:

I am happy with how the authors addressed all the reviewer's concerns.

Author Rebuttal, first revision:

Response to reviewers

Reviewer #1 (Remarks to the Author):

line 217) 'To investigate this, we conducted a second experiment on the same 35 strains were exposed to varying numbers of predators (control = 0 per mL, low = ca. 15 individuals mL⁻¹ and high = ca. 50219 individuals mL⁻¹). ' (sentence reads weirdly)

We have now rephrased this sentence, which now reads “To investigate this, we conducted a second experiment where the same 35 strains were exposed to different densities of predators”

line 248) "mother cells divide into four daughter cells " consider adding "mitotically" to avoid confusion with meiotic cell division”

We have added mitotically to the sentence.

Final Decision Letter:

22nd March 2023

Dear Charlie,

We are pleased to inform you that your Article entitled "Single cell adaptations shape evolutionary transitions to multicellularity in green algae", has now been accepted for publication in Nature Ecology & Evolution.

21Over the next few weeks, your paper will be copyedited to ensure that it conforms to Nature Ecology and Evolution style. Once your paper is typeset, you will receive an email with a link to choose the appropriate publishing options for your paper and our Author Services team will be in touch regarding any additional information that may be required

You will not receive your proofs until the publishing agreement has been received through our system

Due to the importance of these deadlines, we ask you please us know now whether you will be difficult to contact over the next month. If this is the case, we ask you provide us with the contact information (email, phone and fax) of someone who will be able to check the proofs on your behalf, and who will be available to address any last-minute problems . Once your paper has been scheduled for online publication, the Nature press office will be in touch to confirm the details.

Acceptance of your manuscript is conditional on all authors' agreement with our publication policies (see www.nature.com/authors/policies/index.html). In particular your manuscript must not be published elsewhere and there must be no announcement of the work to any media outlet until the publication date (the day on which it is uploaded onto our web site).

Please note that *Nature Ecology & Evolution* is a Transformative Journal (TJ). Authors may publish their research with us through the traditional subscription access route or make their paper immediately open access through payment of an article-processing charge (APC). Authors will not be required to make a final decision about access to their article until it has been accepted. [Find out more about Transformative Journals](https://www.springernature.com/gp/open-research/transformative-journals)

Authors may need to take specific actions to achieve [compliance with funder and institutional open access mandates](https://www.springernature.com/gp/open-research/funding/policy-compliance-faqs). If your research is supported by a funder that requires immediate open access (e.g. according to [Plan S principles](https://www.springernature.com/gp/open-research/plan-s-compliance)) then you should select the gold OA route, and we will direct you to the compliant route where possible. For authors selecting the subscription publication route, the journal's standard licensing terms will need to be accepted, including [self-archiving-and-license-to-publish](https://www.nature.com/nature-portfolio/editorial-policies/self-archiving-and-license-to-publish). Those licensing terms will supersede any other terms that the author or any third party may assert apply to any version of the manuscript.

22If you have any questions about our publishing options, costs, Open Access requirements, or our legal forms, please contact ASJournals@springernature.com

We welcome the submission of potential cover material (including a short caption of around 40 words) related to your manuscript; suggestions should be sent to Nature Ecology & Evolution as electronic files (the image should be 300 dpi at 210 x 297 mm in either TIFF or JPEG format). Please note that such pictures should be selected more for their aesthetic appeal than for their scientific content, and that colour images work better than black and white or grayscale images. Please do not try to design a cover with the Nature Ecology & Evolution logo etc., and please do not submit composites of images related to your work. I am sure you will understand that we cannot make any promise as to whether any of your suggestions might be selected for the cover of the journal.

You can generate the link yourself when you receive your article DOI by entering it here: <http://authors.springernature.com/share>.

[REDACTED]

P.S. Click on the following link if you would like to recommend Nature Ecology & Evolution to your librarian <http://www.nature.com/subscriptions/recommend.html#forms>

** Visit the Springer Nature Editorial and Publishing website at http://editorial-jobs.springernature.com?utm_source=ejp_NEcoE_email&utm_medium=ejp_NEcoE_email&utm_campaign=ejp_NEcoE for more information about our career opportunities. If you have any questions please click [here](mailto:editorial.publishing.jobs@springernature.com).**